# Computational modeling of spinal circuits controlling limb coordination and gaits in quadrupeds

Simon M Danner[1]*, Natalia A Shevtsova[1], Alain Frigon[2], Ilya A Rybak[1]

[1]Department of Neurobiology and Anatomy, Drexel University College of Medicine, Philadelphia, United States; [2]Department of Pharmacology-Physiology, Université de Sherbrooke, Sherbrooke, Canada

**Abstract** Interactions between cervical and lumbar spinal circuits are mediated by long propriospinal neurons (LPNs). Ablation of descending LPNs in mice disturbs left-right coordination at high speeds without affecting fore-hind alternation. We developed a computational model of spinal circuits consisting of four rhythm generators coupled by commissural interneurons (CINs), providing left-right interactions, and LPNs, mediating homolateral and diagonal interactions. The proposed CIN and diagonal LPN connections contribute to speed-dependent gait transition from walk, to trot, and then to gallop and bound; the homolateral LPN connections ensure fore-hind alternation in all gaits. The model reproduces speed-dependent gait expression in intact and genetically transformed mice and the disruption of hindlimb coordination following ablation of descending LPNs. Inputs to CINs and LPNs can affect interlimb coordination and change gait independent of speed. We suggest that these interneurons represent the main targets for supraspinal and sensory afferent signals adjusting gait.

DOI: https://doi.org/10.7554/eLife.31050.001

*For correspondence:
simon.danner@gmail.com

## Introduction

To effectively move in the environment, limbed animals use different gaits depending on the desired locomotor speed (*Grillner, 1981*; *Hildebrand, 1989*). Mice change gait from walk to trot followed by gallop and bound as locomotor speed increases (*Clarke and Still, 1999*; *Herbin et al., 2004*; *Herbin et al., 2007*; *Bellardita and Kiehn, 2015*; *Lemieux et al., 2016*). Each gait is characterized by a distinct pattern of limb movements (*Hildebrand, 1989*), which in turn is defined by neural interactions between spinal circuits controlling each limb.

It is generally accepted that each limb is controlled by a separate central pattern generator (CPGs), because cats are able to walk on split-belt treadmills with limbs stepping at different speeds (*Forssberg et al., 1980*; *Halbertsma, 1983*; *Frigon et al., 2013, 2015*; *Thibaudier et al., 2013*; *Frigon, 2017*). Hence, quadrupedal locomotion is controlled by four CPGs (one per limb) located in separate sections of the spinal cord, specifically on the left and right side in the lumbar and cervical enlargements (*Kato, 1990*; *Ballion et al., 2001*; *Juvin et al., 2005*; *Juvin et al., 2012*). Therefore, the central control of interlimb coordination and gait expression is defined by neurons and neuronal pathways that mediate interactions between circuits controlling each limb. These interlimb pathways are mediated by (a) short projecting commissural interneurons (CINs) that control left-right interactions at the lumbar and, presumably, cervical levels (*Talpalar et al., 2013*; *Bellardita and Kiehn, 2015*; *Molkov et al., 2015*; *Shevtsova et al., 2015*; *Danner et al., 2016*), and (b) descending (cervical-to-lumbar) and ascending (lumbar-to-cervical) long propriospinal neurons (LPNs) that control homolateral and diagonal cervical-lumbar interactions (*Matsushita et al., 1979*; *Skinner et al.,*

*1979*, *1980*; *Menétrey et al., 1985*; *Nathan et al., 1996*; *Dutton et al., 2006*; *Brockett et al., 2013*; *Ruder et al., 2016*; *Flynn et al., 2017*).

The role and speed-dependent involvement of genetically identified CINs, specifically V0$_V$ and V0$_D$, in alternation and expression of different gaits have been experimentally investigated (*Talpalar et al., 2013*; *Bellardita and Kiehn, 2015*). Specifically, it was shown that mutants lacking V0$_V$ CINs selectively lose trot, and mutants lacking both V0$_V$ and V0$_D$ CINs can only bound at any speed (*Bellardita and Kiehn, 2015*; *Kiehn, 2016*). Using modeling (*Danner et al., 2016*), we presented a possible organization of the central pathways mediating interlimb coordination that exhibited speed-dependent gait expression and the correct loss of gaits after selective removal of V0$_V$ and both V0$_V$ and V0$_D$ CINs. These V0 interneurons may contain both short projecting CINs as well as diagonally projecting LPNs (*Ruder et al., 2016*).

LPNs that directly couple the cervical and lumbar segments have been anatomically identified (*Edinger, 1896*; *Nathan and Smith, 1959*; *Matsushita et al., 1979*; *Skinner et al., 1979*; *Skinner et al., 1980*; *Menétrey et al., 1985*; *Nathan et al., 1996*; *Dutton et al., 2006*; *Brockett et al., 2013*; *Flynn et al., 2017*). Several studies *in vitro* found that LPNs can be involved in coordination of left-right and cervical-lumbar activities recorded from four ventral roots (*Ballion et al., 2001*; *Juvin et al., 2005*; *Juvin et al., 2012*; *Cowley et al., 2010*). However, their role in speed-dependent gait expression *in vivo*, remains poorly understood. Recently, *Ruder et al. (2016)* described axonal projections of different LPNs and their synaptic and genetic properties. These data differ from the connections proposed in our previous model (*Danner et al., 2016*). In addition, *Ruder et al. (2016)* have shown that deletion of descending (cervical-to-lumbar) LPNs causes a speed-dependent distortion of left-right coordination. These new findings provided us with the information needed to revise our model and investigate the role of different LPN pathways in interlimb coordination and gait control during locomotion.

In locomotion induced by stimulation of the mesencephalic locomotor region (MLR), increasing the intensity of stimulation causes an increase of locomotor speed accompanied by sequential gait transitions (*Orlovskiĭ et al., 1966*; *Shik et al., 1966*; *Shik and Orlovsky, 1976*; *Skinner and Garcia-Rill, 1984*; *Grillner, 1985*; *Atsuta et al., 1990*; *Nicolopoulos-Stournaras and Iles, 2009*). From these observations, we hypothesized that the same brainstem drive that controls speed by exciting the rhythm generators also affects interlimb coordination by acting on CINs or LPNs (*Danner et al., 2016*). This hypothesis is in accordance with multiple data confirming that different CINs and LPNs receive supraspinal inputs (*Lloyd, 1942*; *Lloyd and McIntyre, 1948*; *Bannatyne et al., 2003*; *Jankowska et al., 2003*; *Cowley et al., 2010*; *Zaporozhets et al., 2011*). However, the involvement of these neurons in mediating brainstem control of gait has not been demonstrated. Therefore, organization and role of brainstem inputs to these neurons can so far only be based on predictions derived from computational models.

Here, we present a computational model of the spinal locomotor circuits and investigate the potential role of CINs and LPNs in the control of and transition between different locomotor gaits. The model reproduces the speed-dependent gait expression in intact and genetically transformed mice, lacking V2a, V0$_V$, and all V0 neurons (*Crone et al., 2009*; *Bellardita and Kiehn, 2015*), as well as the disruption of hindlimb

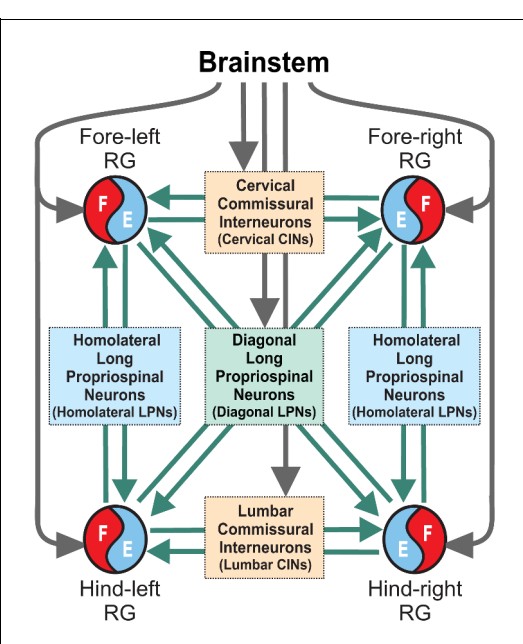

**Figure 1.** Model concept. Each limb is controlled by a separate rhythm generator (RG). The local commissural neurons (CINs) as well as homolaterally and diagonally projecting (descending and ascending) long propriospinal neurons (LPNs) couple the four RGs. Brainstem drive acts on the RGs to control locomotor speed and on CINs and diagonal LPNs to control gaits.
DOI: https://doi.org/10.7554/eLife.31050.002

coordination following ablation of descending LPNs (*Ruder et al., 2016*). The model proposes the following roles for LPNs: diagonal V0$_D$ LPNs supports walk, diagonal V0$_V$ LPNs together with local V0$_V$ CINs stabilize trot, and homolateral LPNs ensure fore-hind alternation in all gaits. Finally, we show that additional external inputs to CINs and LPNs may affect interlimb coordination and gait expression independent of speed. We suggest that these spinal interneurons are the main targets of supraspinal and somatosensory afferents to adjust interlimb coordination and gait to different environmental and behavioral conditions.

## Results

### The model

The model consists of four rhythm generators (RGs), each controlling one limb (*Figure 1*). These RGs interact via several bidirectional pathways mediated by CINs (left-right) and LPNs (homolateral and diagonal). Neural populations were modeled with a simplified, 'activity-dependent' population model (see Materials and methods). Each RG consisted of a flexor (F) and extensor center (E) that mutually inhibited each other through intermediate inhibitory interneurons (InF and InE, respectively). Similar to our previous models (*Rybak et al., 2006a*, *2006b*, *2013*, *2015*; *Molkov et al., 2015*; *Shevtsova et al., 2015*; *Danner et al., 2016*), both centers incorporated a slowly inactivating, persistent sodium current ($I_{NaP}$), allowing each center to intrinsically generate rhythmic bursting under certain conditions defined by external tonic drives or level of excitation, which was consistent with experimental data (*Hägglund et al., 2013*). However, we followed the concept of asymmetric flexion-dominated CPG organization (*Pearson and Duysens, 1976*; *Duysens, 1977*; *Zhong et al., 2012*; *Duysens et al., 2013*; *Machado et al., 2015*). Based on this concept and our previous models (*Rybak et al., 2015*; *Molkov et al., 2015*; *Shevtsova et al., 2015*; *Danner et al., 2016*; *Ausborn et al., 2017*), we assumed that under normal conditions extensor centers receive relatively high drive that keeps them in the regime of tonic activity. Hence, they exhibited rhythmic bursting only due to rhythmic inhibition from the corresponding intrinsically oscillating flexor centers. In this case, the frequency of oscillation in the model was defined by the brainstem drive to flexor centers and was almost independent of the phasic interactions between RGs mediated by CINs and LPNs (*Rybak et al., 2015*).

Left-right CIN connections at cervical and lumbar levels (*Figure 2A* and *Figure 3*) are organized in accordance to our previous models (*Rybak et al., 2015*; *Shevtsova et al., 2015*; *Danner et al., 2016*): the inhibitory V0$_D$ CINs provide direct mutual inhibition between the left and right flexor centers, the excitatory V0$_V$ CINs also provide mutual inhibition between the flexor centers (receiving inputs from excitatory V2a and acting through inhibitory Ini neurons), the excitatory V3 CINs provide mutual excitation between the flexor centers, and the inhibitory CINi CINs provide inhibition from the extensor to the contralateral flexor centers.

Connections between the cervical and lumbar circuits are mediated through mono- and polysynaptic neural pathways. Here, we modeled them as LPN connections. *Ruder et al. (2016)* reported that there are excitatory and inhibitory descending LPNs, whereas ascending LPNs are only excitatory. Both descending and ascending LPNs project homolaterally and diagonally. There are more diagonal then homolateral excitatory LPNs, while the opposite is the case for inhibitory LPNs. The excitatory homolateral LPNs were identified as Shox2 and the excitatory diagonal LPNs as V0$_V$ neurons. We have hypothesized that diagonally projecting inhibitory LPNs are of V0$_D$ type.

The cervical-to-lumbar connections in our model (*Figure 2B*) are organized accordingly: the excitatory Shox2 neurons provide excitation from each cervical extensor to its homolateral lumbar flexor center, inhibitory LPNs of unidentified type (LPNi) provide inhibition from each cervical flexor to its homolateral lumbar flexor center, V0$_V$ LPNs provide excitation from each cervical flexor to its diagonal flexor center, and inhibitory V0$_D$ LPNs provide inhibition from each cervical flexor to its diagonal lumbar flexor center.

Lumbar-to-cervical connections (*Figure 2C*) mirror the excitatory cervical-to-lumbar connections but do not include inhibitory LPNs.

Finally, there were two brainstem drives (*Figure 2D*): one that excited flexor centers and inhibited the local V0 CINs at cervical and lumbar levels and the descending diagonal V0$_D$ LPNs, promoting left-right and diagonal alternation between the corresponding RGs, and the other that provided

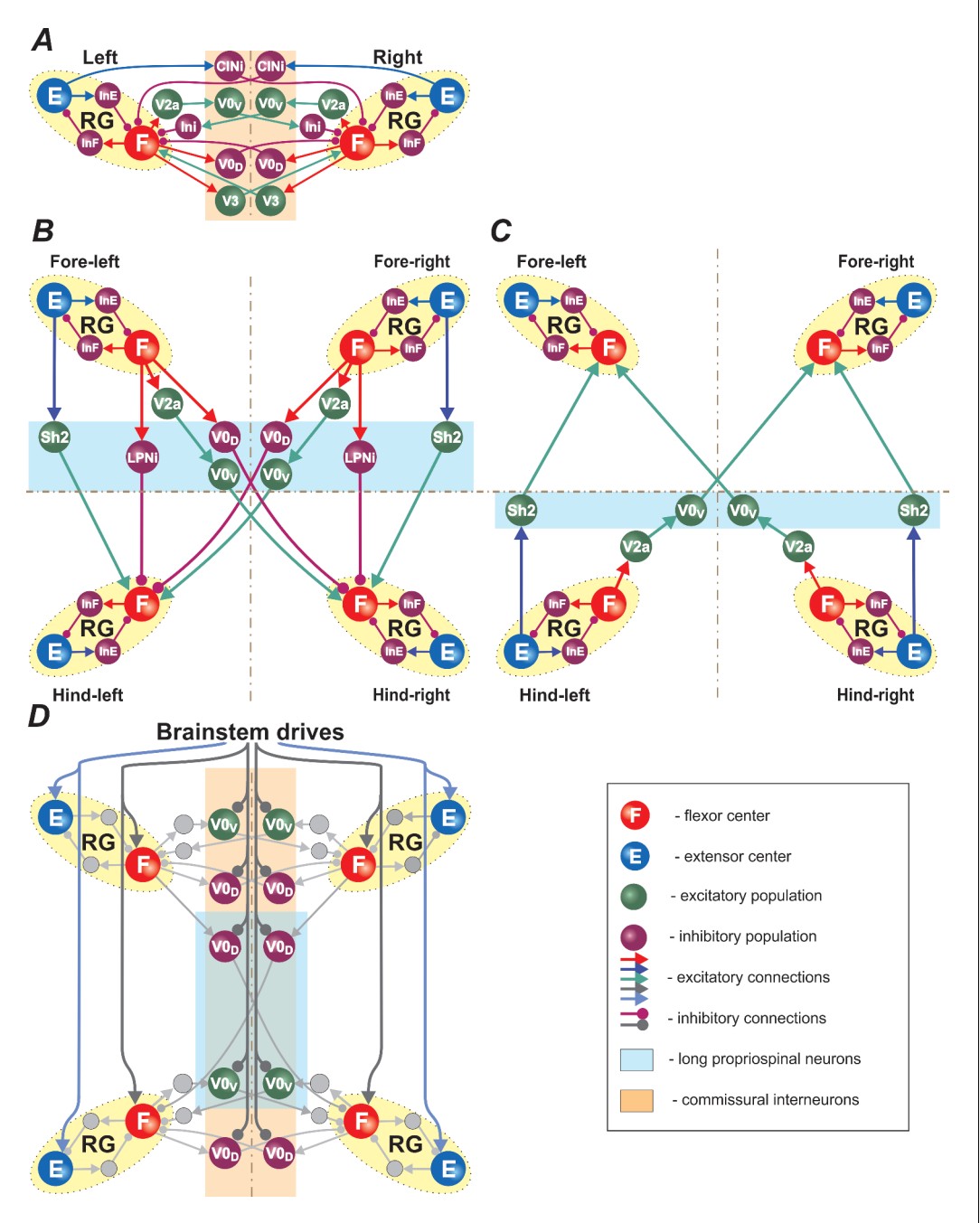

**Figure 2.** Connections within the spinal cord. (**A**) Connections between left and right rhythm generators (RG) within each girdle. (**B**) Connections from the fore to hind RGs via descending (cervical-to-lumbar) long propriospinal neurons (LPNs). (**C**) Connections from the hind to fore RGs via ascending (lumbar-to-cervical) LPNs. (**D**) Brainstem drive connections to the extensor and flexor centers, commissural interneurons, and LPNs. Spheres represent neural populations. Excitatory and inhibitory connections are marked by arrowheads and circles, respectively.

DOI: https://doi.org/10.7554/eLife.31050.003

constant excitatory drive to the extensor centers. The drive to the flexor centers, CINs, and LPNs was varied to influence locomotor speed and gait. Specifically, the drive to flexor centers defined the locomotor frequency. Based on our suggestion, this drive also inhibited V0 CINs and diagonal

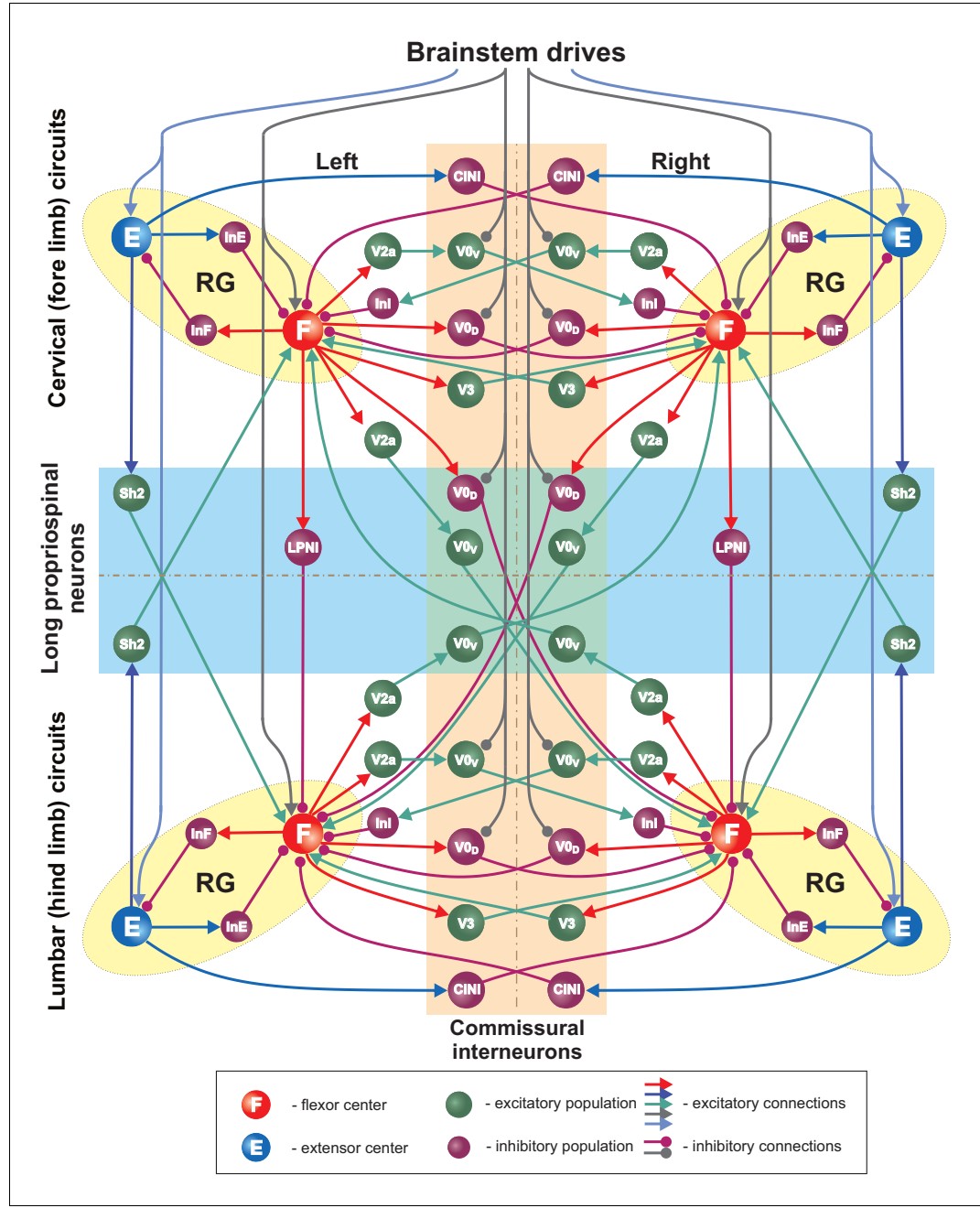

**Figure 3.** Full model schematic. Spheres represent neural populations and lines synaptic connections. Excitatory and inhibitory connections are marked by arrowheads and circles, respectively. RG, rhythm generator.
DOI: https://doi.org/10.7554/eLife.31050.004

$V0_D$ LPNs changing the balance of excitatory and inhibitory interactions between the corresponding RGs. The full model schematic can be seen in *Figure 3*.

The model was used to analyze drive- and frequency-dependent changes in the locomotor phase durations and the expression of different gaits in an intact system and following the removal of particular neuron types.

## Control of locomotor frequency and gait by brainstem drive

We investigated the model performance by changing the parameter $\alpha$, which defined brainstem drive to flexor centers, CINs, and LPNs (see Materials and methods). The model generated

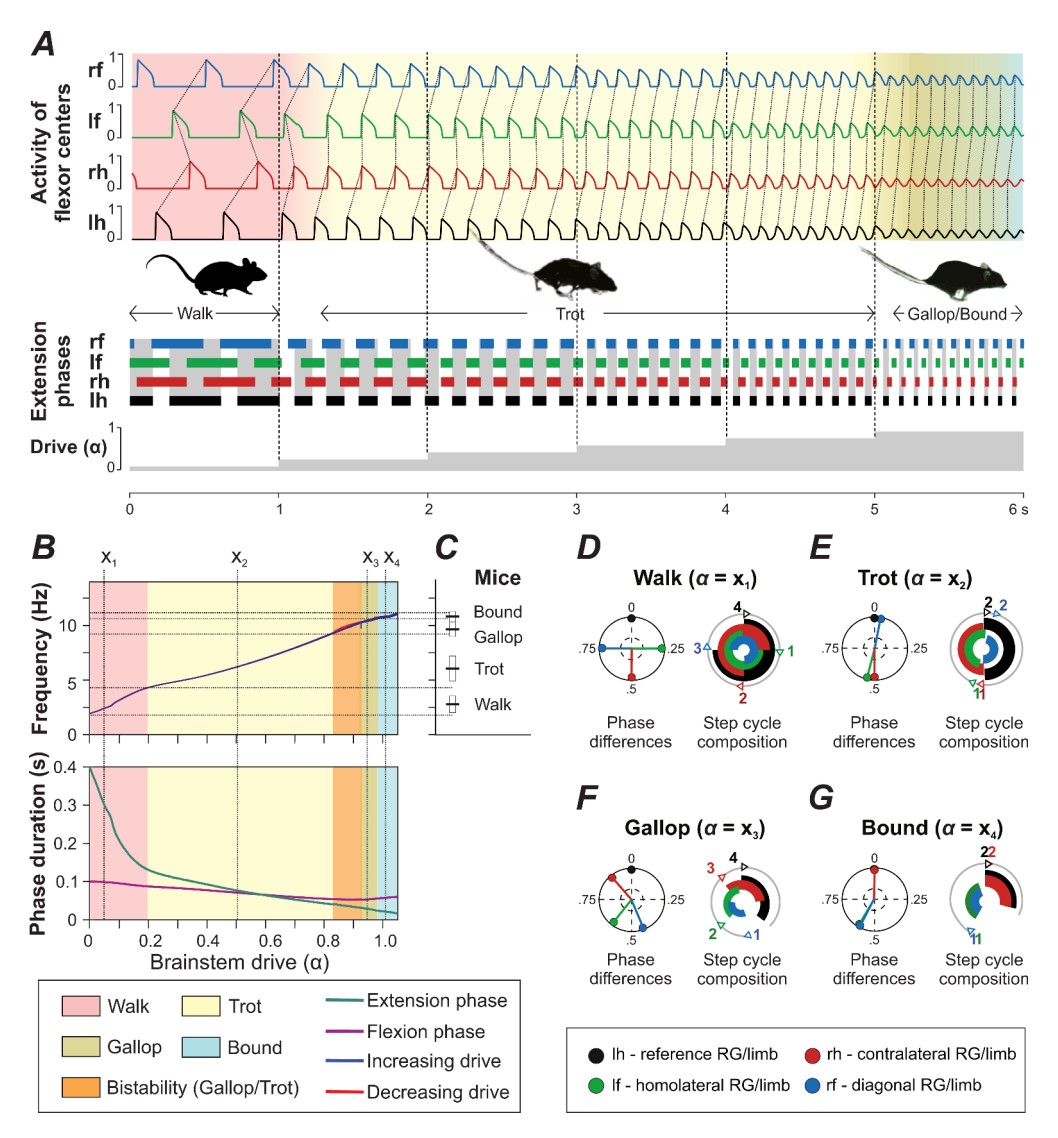

**Figure 4.** Model performance. (**A**) Gait changes during 6 s of simulation with increase of brainstem drive (parameter $\alpha$) every second (shown at the bottom). Gait changes were subject to a brief transitional period (shown as transitional colors). (**B**) Dependency of frequency and phase durations on $\alpha$. Vertical dashed lines (labeled as $x_1 - x_4$) represent selected values of $\alpha$ related to different gaits. Note that the blue line almost completely covers the red one (indicating increase and decrease of $\alpha$, respectively) in the top diagram. (**C**) Frequency ranges in which gaits are observed in mice (created from data from Figure 1E of **Bellardita and Kiehn, 2015**). (**D–G**) Phase differences and step cycle compositions of four characteristic gaits produced by the model that correspond to the $\alpha$-values ($x_1 - x_4$) in B.

DOI: https://doi.org/10.7554/eLife.31050.005

oscillations when parameter $\alpha$ was changed from 0 to 1.05. Within this range, an increase of $\alpha$ led to an increase in locomotor frequency from 2 to 12 Hz (**Figure 4A and B**, top diagram). This increase in frequency occurred mostly due to shortening of the extension phase, while the flexion phase remained almost constant (**Figure 4B**, bottom diagram), which is a characteristic property of fictive and real locomotion across species (**Halbertsma, 1983**; **Hildebrand, 1989**; **Frigon and Gossard, 2009**; **Danner et al., 2015**; **Shevtsova et al., 2015**). The increased frequency in the model was accompanied by sequential gait changes from walk to trot to gallop and then bound (**Figure 4A**) consistent with experimental observations (**Figure 4C**, **Bellardita and Kiehn, 2015**).

The expressed walking gait was a lateral-sequence walk (*Figure 4A,D*), in which flexion phases proceeded in the following order: hind-right, fore-right, hind-left and fore-left. This walk occurred at frequencies from 2 to 4 Hz (*Figure 4B*). At low frequencies, the extension phase was three to four times longer than the flexion phase duration, the RGs were in flexion sequentially, and the homolateral and diagonal phase differences were close to 0.25 and 0.75, respectively (*Figure 4D*).

Trot (*Figure 4A,E*) was characterized by diagonal synchronization, left-right alternation and fore-hind alternation. It was expressed over a large range of frequencies (from 4 to 10.5 Hz; *Figure 4B*). At low frequencies, with longer extension than flexion phase durations, the gait constituted a walking trot and otherwise a running trot.

Gallops are characterized by synchronization or quasi-synchronization of the hind RGs and a non-zero phase difference between the fore RGs. In the model (*Figure 4A,B,F*), gallop was expressed at frequencies between 9 and 11 Hz (between 9 and 10.5 Hz both gallop and trot occurred). The sequence of flexion phases at cervical or lumbar levels could be either left-right or right-left but was always the same for both fore and hind RGs. Thus, transverse but not rotary gallops were expressed. The left-right phase difference of the hind RGs was always closer to synchronization than of the fore RGs. Some of the gallops exhibited an interval between the end of the hind extension and the beginning of the fore extension, in which all RGs were in flexion at the same time (*Figure 4F*). This corresponds to an aerial phase, characteristic for gallops.

Bound (*Figure 4A,B,G*) was characterized by left-right synchronization at cervical and lumbar levels, and homolateral and diagonal alternation. The fore extension phases were directly followed by hind extension phases, which were then followed by all RGs being in flexion (corresponding to an aerial phase). Bound was expressed at frequencies between 11 and 12 Hz. The flexion phase duration was always longer than the extension phase duration (*Figure 4B*, bottom diagram).

The frequency ranges, phase differences, and step cycle compositions (relative phase durations) of all gaits were consistent with mouse locomotion (*Figure 4B–G*; *Bellardita and Kiehn, 2015*; see gait definitions in Materials and methods).

## Walk and the transition to trot: diagonal V0$_D$ LPNs ensure stable walk

To investigate control of locomotion and gait transitions by brainstem drive, we used bifurcation diagrams reflecting normalized phase differences between oscillations in hind and fore left-right, homolateral and diagonal RGs with changing parameter $\alpha$ that represented the brainstem drive and was used as bifurcation parameters (see examples in *Figure 5*). Normalized phase differences of 0.5 correspond to alternation, whereas phase differences of 0 or 1 correspond to synchronization. Gaits were operationally defined based on these phase differences (see Table 2 in Materials and methods) and are marked in the diagrams by colored areas. Blue and red lines indicate the stable phase differences with stepwise increase and decrease of $\alpha$, respectively. Any discrepancies between the red and blue lines indicate regions of bi- or multistability. For these $\alpha$ values two or more different stable gaits coexisted and could be expressed depending on the initial conditions. Bifurcations can be seen as abrupt changes of the stable phase differences. *Figure 5A* shows the bifurcation diagrams of the intact model.

At the lowest values of brainstem drive, both left-right (local V0$_D$ and V0$_V$ CINs) and diagonal alternation (diagonal V0$_D$ LPNs) promoting pathways dominated. Together with three to four times longer extension than flexion phase durations (*Figure 4B* bottom diagram) and homolateral alternation (present over all frequencies), this ensured that only one RG was in flexion at any given time (*Figure 4A,D* and *Figure 5A*).

With increasing $\alpha$, inhibitory brainstem drive caused the activity of the diagonally projecting V0$_D$ LPNs to decrease and the balance to shift from diagonal alternation (promoted by V0$_D$) to diagonal synchronization (promoted by V0$_V$). Together with a progressive decrease in the relative duration of the extension phase, this caused the flexion phases of the diagonal RGs to progressively overlap and the diagonal phase difference to move from a quarter-phase lag toward synchronization (*Figure 5A*). In parallel, the homolateral phase difference moved toward alternation and the left-right phase differences remained at a strict out-of-phase alternation (0.5 normalized phase difference). Together this constitutes a gradual transition from walk to trot.

The sequence of the flexion phases during walk was determined by the hypothetical CINi population that was introduced in our previous model (CINi2 in *Danner et al., 2016*) to ensure 0.5 normalized left-right phase difference during walk.

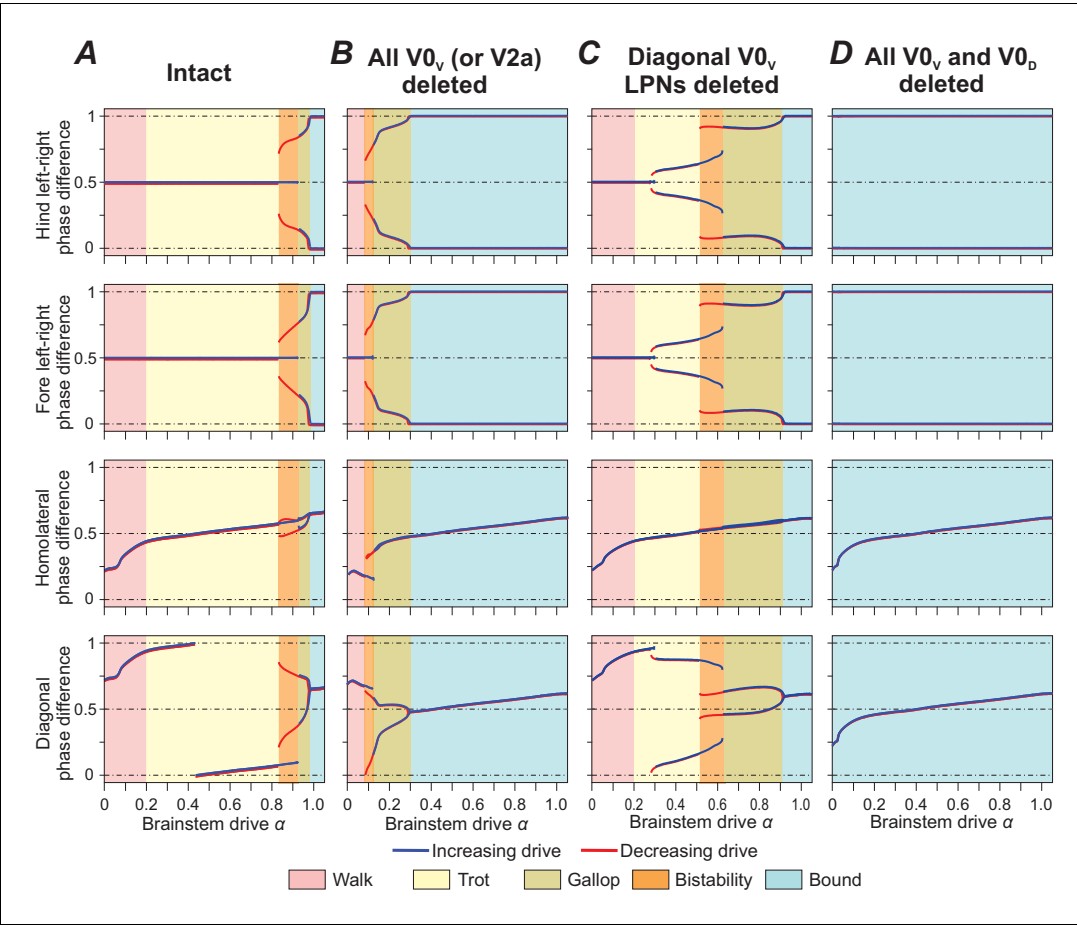

**Figure 5.** Bifurcation diagrams of the intact model (**A**), after removal of all $V0_V$ neurons (**B**), of only $V0_V$ LPNs (**C**), and of all $V0_V$ and $V0_D$ neurons (**D**). Normalized phase differences of 0.5 correspond to alternation, whereas phase differences of 0 or 1 correspond to synchronization. Blue and red lines indicate the stable phase differences with stepwise increase and decrease of the bifurcation parameter $\alpha$, respectively. Colored areas indicate the expressed gait.

DOI: https://doi.org/10.7554/eLife.31050.006

## Trot and the transition to gallop and bound: diagonal $V0_V$ LPNs stabilize trot

The transition from trot to gallop and bound was governed by brainstem-drive-controlled speed-dependent changes in the left-right and diagonal pathways. The left-right alternation promoting local CINs ($V0_D$ and $V0_V$) received inhibition from the brainstem that increased with speed, while the left-right synchronization promoting local CINs (V3 and CINi) did not receive brainstem inhibition (*Figure 2D*, *Figure 3*). Thus, with increasing brainstem drive (and frequency), the pathway promoting left-right synchronization became stronger than those promoting left-right alternation. This resulted in a transition from trot to gallop and then to bound (*Figure 5A*). The diagonally projecting $V0_V$ LPNs, promoting diagonal synchronization, acted synergistically with local $V0_V$ CINs.

Two sequential bifurcations occurred: when the gait switched from trot to gallop and then from gallop to bound (*Figure 5A*). The bifurcation from trot to gallop resembled a subcritical (backwards) pitchfork bifurcation in the hind (and fore) left-right phase differences. The transition exhibited hysteresis: perfect left-right alternation and quarter-phase lags (in either direction) were stable at the same $\alpha$-values (blue and red lines show different stable left-right phase differences in the orange shaded area in *Figure 5A*). In other words, when drive was increasing (blue lines), the transition from trot to gallop occurred at a higher $\alpha$-value than the transition from gallop to trot when drive was decreasing (red lines). Such a hysteresis is a common feature of gait transitions in quadrupeds

(*Heglund and Taylor, 1988*). Stable gallops occurred because the decreasing $V0_V$ CINs and LPNs activities were accompanied by a decrease of the burst duration, while V3 and CINi CINs were active during the whole flexion or extension phase, respectively. Note, that the previous model (*Danner et al., 2016*) could not reproduce stable gallops; they only existed as a transitional regime. Therefore, the addition of diagonal LPNs in the present model was critical for stable gallop to exist. The bifurcation from trot to gallop occurred when $V0_V$ CINs and LPNs could not support left-right alternation and diagonal synchronization any more, but V3 and CINi CINs were not strong enough to cause perfect left-right synchronization. Therefore, stable states emerged around 0/1 normalized left-right phase differences. The deviation from synchronization was determined by the duration of the bursts of $V0_V$ CINs and LPNs. As the burst durations decreased further, the left-right phase differences transitioned towards synchronization (and the diagonal phase difference towards alternation) via a supercritical pitchfork bifurcation. This bifurcation provided the transition from gallop to bound without hysteresis (blue and red lines of the left-right phase differences in *Figure 5A* follow the same trajectory).

## Gait expression following removal of different spinal interneuron types

Experimental observations showed that mice lacking $V0_V$ neurons lose trot, mice lacking $V0_V$ and $V0_D$ only exhibit bound (*Bellardita and Kiehn, 2015*), and mice lacking V2a neurons transition to left-right synchronization at lower speeds compared to intact animals (*Crone et al., 2009*). Thus, to further validate the model, we removed several types of genetically identified neurons and compared the model performance with experimental results.

When all $V0_V$ neurons were deleted (*Figure 5B*) only walk, gallop and bound were expressed; trot was selectively lost. Left-right synchronization promoting pathways became dominant at a lower $\alpha$-value (lower frequency), and left-right alternation was only supported by $V0_D$ CINs. Deletion of V2a had the same effect as deletion of $V0_V$, because V2a relayed all inputs to $V0_V$ CINs and LPNs.

Deletion of only diagonally projecting $V0_V$ LPNs (and not $V0_V$ CINs) did not result in a loss of any gait, but caused the transition from trot to gallop to occur at a lower $\alpha$-value and frequency (*Figure 5C*). Furthermore, an additional bifurcation in the left-right and diagonal phase differences occurred during trot. At low trot-frequencies and $\alpha$-values, a strict out-of-phase left-right alternation (0.5 normalized phase difference) was stable, while at medium trot-frequencies, a pair of stable states near 0.5 normalized phase difference emerged that then transitioned via a region of multi-stability to a pair of stable states near 0/1 normalized phase difference (expression of gallop). This highlights how local $V0_V$ CINs and LPNs synergistically stabilize trot by providing left-right alternation and diagonal synchronization.

Deletion of both $V0_V$ and $V0_D$ resulted in the removal of all left-right alternation promoting pathways and caused the loss of all alternating gaits (*Figure 5D*). Bound was stable over all frequencies.

## Deletion of descending (cervical-to-lumbar) LPNs affects left-right coordination

*Ruder et al. (2016)* showed that deletion of descending LPNs influences left-right coordination: at low speeds both the fore and hind limbs were alternating, then at medium speeds the hind limbs, and at high speeds both the fore and hind limbs demonstrated disturbed coordination with both alternating and synchronized activities.

In our model, deletion of the cervical-to-lumbar LPNs resulted in the emergence of new stable states (*Figure 6B*). At low brainstem drive values walk remained the only stable state and with increasing drive it gradually transitioned towards trot. With further increase of $\alpha$, trot was first the only stable gait (*Figure 6B,C*). Then, both trot and gallop were stable at the same $\alpha$-values (*Figure 6B,D1,D2,E1,E2*) and finally bound became the single stable state (*Figure 6B,F*). During the gallop at low $\alpha$-values, the hind RGs were close to synchronization while the fore RGs remained alternating (*Figure 6B,D2*). With increasing $\alpha$, the fore left-right phase differences gradually transitioned towards synchronization, thus gallops at higher $\alpha$-values exhibited fore and hind left-right phase differences close to synchronization (*Figure 6B,E2*).

The introduced asymmetry by the deletion of pathways in only one direction (from cervical to lumbar but not from lumbar to cervical) caused bifurcations of higher codimension and resulted in several bi- and multistabilities (*Figure 6B*). Interestingly, the stable states closely approximate the

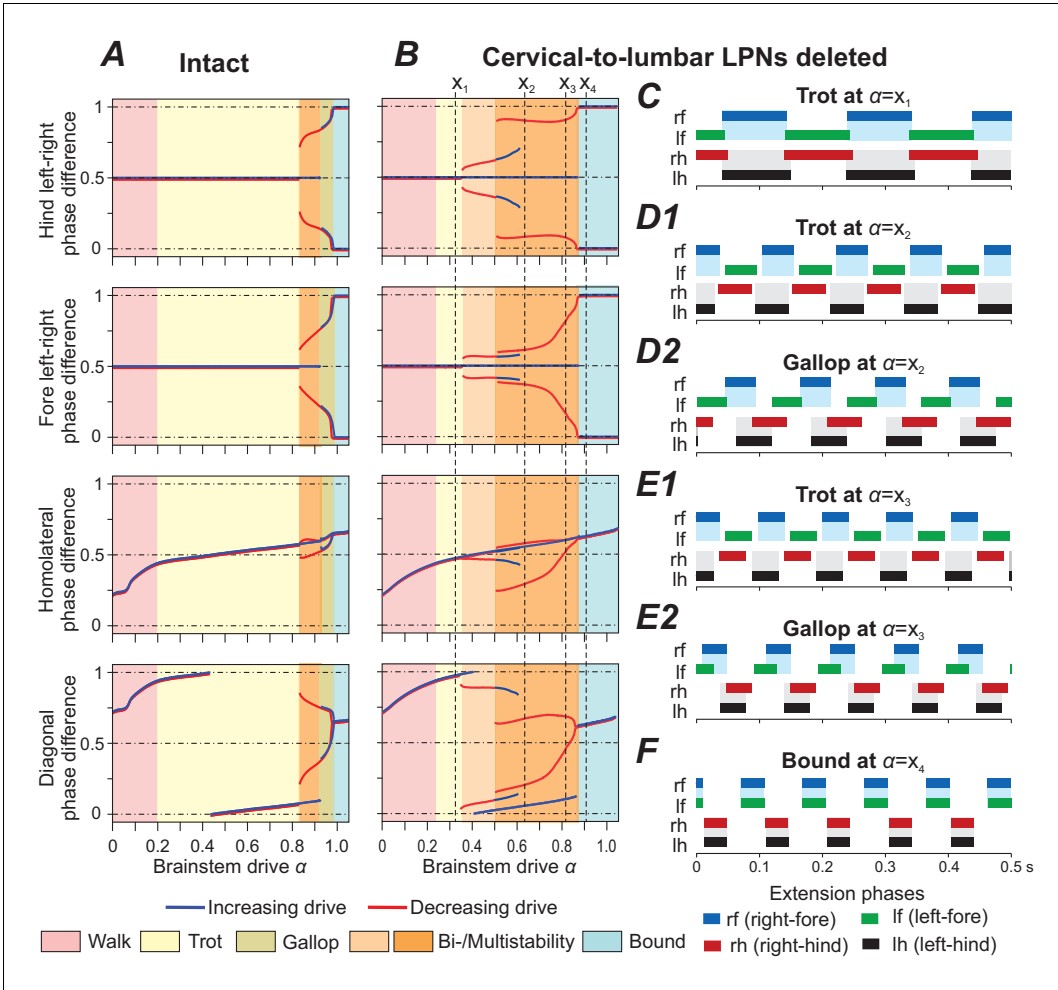

**Figure 6.** Model performance after deletion of all cervical-to-lumbar long propriospinal neurons (LPNs). Bifurcation diagrams, indicating the stable phase differences depending on the bifurcation parameter $\alpha$ for the intact model (**A**) and after deletion of all cervical-to-lumbar LPNs (**B**). (**C–F**) Extension phases of stable gaits (identified by vertical dashed lines in B labeled as $x_1 - x_4$). At $\alpha = \{x_2, x_3\}$ two different gaits were stable simultaneously (depicted in D1, D2 and E1, E2, respectively).
DOI: https://doi.org/10.7554/eLife.31050.007

union of the stable states of the intact model *Figure 6A* and the model after deletion of all diagonally projecting V0$_V$ LPNs (*Figure 5C*). Thus, the disturbance of the fore and hind left-right phase differences was mainly caused by the deletion of diagonally projecting V0$_V$ LPNs. The remaining homolaterally projecting LPNs maintained fore-hind alternation after deletion. Local V0$_V$ CINs maintained the stable branch of left-right alternation.

## Noise causes high step-to-step variability after deletion of cervical-to-lumbar LPNs

In the above section, we described that deletion of the cervical-to-lumbar LPNs resulted in the emergence of new, additional stable states at medium and high brainstem drive values. This means that two gaits were stable at the same $\alpha$-values (e.g. see $x_2$ in *Figure 6B* and *Figure 6D1,D2*). To consider how these changes in the number of steady states affect system behavior, such as step-to-step variability, we increased the noisy currents in all populations ($\sigma_{\text{Noise}}$ from 0.005 pA to 1.75 pA). In the intact model, at medium brainstem drives (related to trot), the increased noise caused phase durations and phase differences to become more variable, but both, the fore and hind RGs remained alternating over all steps (*Figure 7A*). After deletion of cervical-to-lumbar LPNs, the applied noise

caused spontaneous transitions between the steady states: the activity of hind RGs was spontaneously switching between synchronization and alternation while the fore RGs remained alternating (see dashed box in *Figure 7B*).

A similar disruption of interlimb coordination has been described in mice after deletion of cervical-to-lumbar LPNs (*Ruder et al., 2016*). They reported random occurrences of clusters of misaligned steps and spontaneous transitions between alternation and synchronization, which were similar to those exhibited by our model. Furthermore, the model also reproduced the speed-dependent effect of changes in the proportion of appropriate and misaligned steps: at low speeds hind and fore limb coordination remained appropriate for trot, at medium speeds hind limb coordination was disturbed, and at high speeds both fore and hind limb coordination was affected (*Figure 7C–E*). Thus, in the regions where both trot and gallop were stable, the applied noise caused spontaneous transitions between the stable states.

## Gait changes by brainstem drive and by drive-independent inputs to CINs and LPNs

Mice usually switch gaits and locomotor speeds abruptly with few or no intermediary steps (*Bellardita and Kiehn, 2015*). To simulate the dynamics of gait transitions, we initialized the model with a constant $\alpha$-value that defined a certain gait (and frequency) and then abruptly changed $\alpha$ to a value characteristic of another gait. Transitions from walk to trot, from gallop to trot and from walk to gallop and back to walk (*Figure 8A–C*) mirrored experimental observations closely (*Figure 8D–E*; *Bellardita and Kiehn, 2015*): they usually required only one or two cycles to stabilize the new gait and locomotor frequency.

Finally, gait changes could also be induced by additional inputs to CINs and LPNs without changing the common brainstem drive controlling locomotor speed (frequency). For example, additional inhibition to $V0_V$ CINs and LPNs during trot induced a transition to bound without significantly affecting the frequency, resulting in a bound at a frequency characteristic of trot (*Figure 8G*). Excitatory drive to the local $V0_V$ CINs during gallop induced a transition to trot, again without significantly affecting speed (*Figure 8H*). Excitation of the local cervical $V0_V$ CINs during gallop caused the fore left-right phase differences to move farther from synchronization and the left and right extension phases of the fore RGs to overlap less, while the hind RG coordination remained almost unchanged (*Figure 8I*). Thus, tonic inputs to CINs and LPNs were able to alter interlimb coordination and gait almost independently of frequency.

## Discussion

### Control of locomotor speed and speed-dependent gait transitions

Central control of locomotor speed and speed-dependent gait expression involves many neural, biomechanical, metabolic, environmental and behavioral factors. Most of these factors are beyond the scope of the present study. Here, we focused on the potential role of central interactions between rhythm-generating circuits controlling each limb and their regulation by brainstem drives.

In our model, two such drives were considered: constant drive to the extensor centers and variable drive to the flexor centers and various CINs and LPNs. The drive to the extensor centers put them in a tonic activity regime if the flexor centers were not active. This drive may represent a separate descending supraspinal drive to extensor circuits operating through the medial reticulospinal or vestibulospinal tracts and/or inputs from cutaneous afferents and load receptors from extensor muscles (*Hiebert and Pearson, 1999*; *Dietz and Duysens, 2000*; *Bouyer and Rossignol, 2003*; *Rossignol et al., 2006*).

The brainstem drive to the flexor centers put them into a rhythmic bursting regime and strong inhibition between flexor and extensor centers mediated by inhibitory interneurons caused rhythmic flexor-extensor alternation (*Zhang et al., 2014*; *Britz et al., 2015*; *Rybak et al., 2015*; *Shevtsova et al., 2015*; *Danner et al., 2016*; *Shevtsova and Rybak, 2016*). Progressive increase of this drive increased locomotor frequency, in accordance with studies in decerebrate cats and rats, where stimulation of the MLR induced locomotor activity that increased in frequency (speed) with increasing stimulation (*Orlovskiĭ et al., 1966*; *Shik et al., 1966*; *Shik and Orlovsky, 1976*; *Skinner and Garcia-Rill, 1984*; *Grillner, 1985*; *Atsuta et al., 1990*; *Nicolopoulos-Stournaras and*

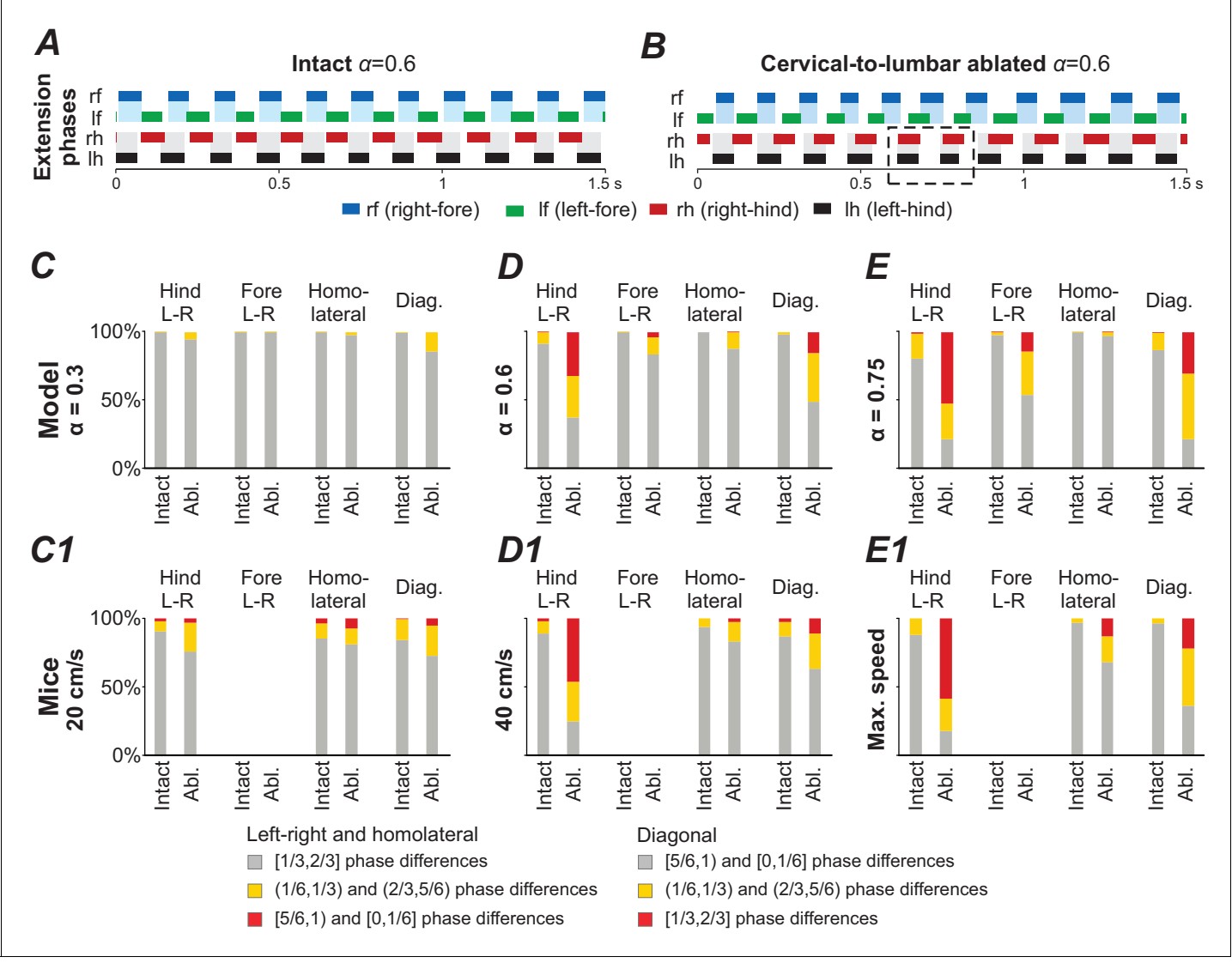

**Figure 7.** Model performance under application of noise before (intact) and after ablation of cervical-to-lumbar long projecting propriospinal neurons (LPNs). (A,B) Exemplary extension phases of the intact model (A) and after removal of cervical-to-lumbar LPNs (B). The dashed box in B indicates synchronization of hind RGs that transiently occurred after removal of cervical-to-lumbar LPNs. (C–E,C1–E1) Phase differences categorized into three equally sized bins (gray: appropriate for trot, yellow: quarter phase difference off, red: antiphase) before and after ablation of cervical-to-lumbar LPNs for 1000 s simulation in the model (C–E) and for experimental data pooled across animals (C1–E1) at three different speeds and $\alpha$-values. C1–E1 were created from data extracted from Figure 5B,C and S5A,D of *Ruder et al. (2016)*.

DOI: https://doi.org/10.7554/eLife.31050.008

*Iles, 2009*). In many of these studies, the increase in speed was accompanied by sequential changes in gait from walk to trot and to gallop and bound. The exact mechanisms for these speed-dependent gait transitions are currently unknown, but it is reasonable to suggest that these transitions are initiated and supported by inputs and drives from different excitatory and inhibitory neurons in the medullary and pontomedullary reticular formation that project to the spinal cord and can directly affect different CINs and LPNs (*Alstermark et al., 1987*; *Jankowska et al., 2003*, *2005*; *Matsuyama et al., 2004*; *Mitchell et al., 2016*; *Ruder et al., 2016*).

Following our previous model (*Danner et al., 2016*), we suggest that gait transitions are caused by brainstem drives acting on neurons coupling the RGs and affecting the balance between the activities of pathways promoting synchronization and alternation. In the previous model, only homolateral LPNs and local CINs were considered. The transition from alternating (trot) to synchronized

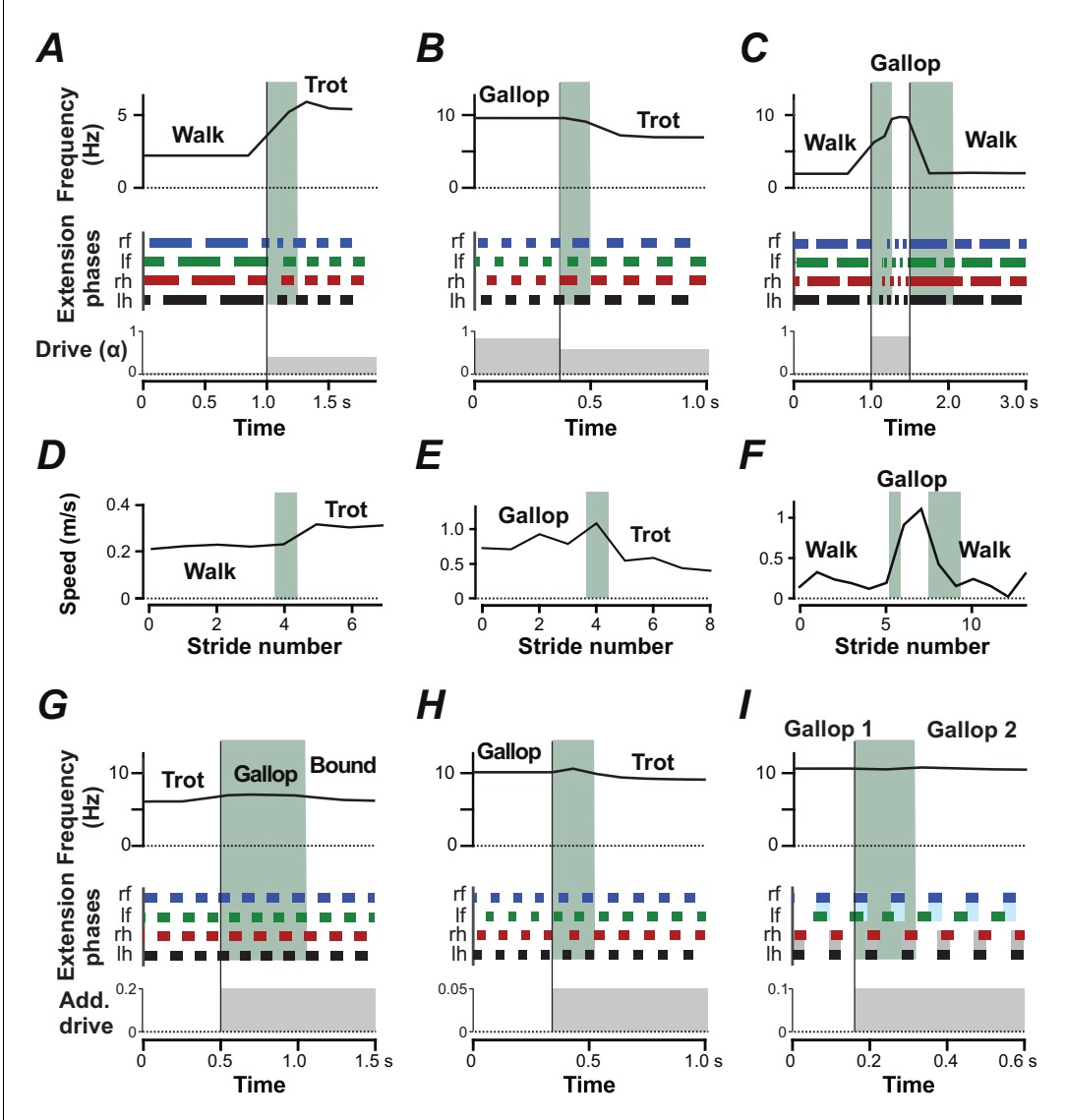

**Figure 8.** Gait transitions with changes of brainstem drive (**A–C**) and independent of brainstem drive (**G–I**). (**A–C**) Instantaneous frequency, extension phases before and after parameter $\alpha$ was abruptly changed (bottom trace). Black vertical lines indicate the time when $\alpha$ was changed and the shaded green areas indicate the transitions periods. (**A**) Transition from walk to trot when $\alpha$ was changed from 0.02 to 0.4. (**B**) Transition from gallop to trot when $\alpha$ was changed from 0.85 to 0.6. (**C**) Transition from walk to gallop and back to walk when $\alpha$ was changed from 0.02 to 0.9 and to 0.02. (**D–F**) Experimentally observed instantaneous speed during gait changes in mice corresponding to A–C (created from data extracted from Figure 3A–C of *Bellardita and Kiehn 2015*). (**G–I**) Instantaneous frequency and extension phases during gait changes caused by additional drives to CINs and LPNs. Bottom trace indicates time course of additional drives. (**G**) Additional inhibitory drive to all V0$_V$ CINs and LPNs ($m_I = 0.0$, $b_I = 0.2$) at $\alpha = 0.5$ and caused a transition from trot to bound. During the transitional period a gallop occurred. (**H**) An additional excitatory drive to local V0$_V$ CINs ($m_E = 0.0$, $b_E = 0.05$) at $\alpha = 0.925$ caused a transition from gallop to trot. (**I**) Additional excitatory drive to cervical, local V0$_V$ CINs ($m_E = 0.0$, $b_E = 0.1$) at $\alpha = 0.975$ caused the left-right phase difference of the fore RGs to change from almost synchronized to a quarter-phase lag during gallop.

DOI: https://doi.org/10.7554/eLife.31050.009

gaits (gallop and bound) resulted from excitatory drive to local V3 CINs, promoting left-right synchronization in both cervical and lumbar levels, so that with increasing drive the left-right interactions through these pathways overcame interactions via pathways supporting left-right alternation.

In the current model, the same conceptual idea was implemented by increasing inhibitory influence of the brainstem drive to local V0 CINs (V0$_D$ and V0$_V$) and diagonal V0$_D$ LPNs (*Figure 2D* and *Figure 3*). This increasing inhibition influenced both the walk to trot and the trot to gallop and bound transitions: the walk to trot transition occurred when diagonal V0$_V$ became stronger than diagonal V0$_D$ LPNs, and the trot to gallop and bound transition occurred when local V3 became stronger than V0$_V$ CINs and LPNs. Additionally, this inhibitory drive allowed for stable solutions for gallop, which in the previous model only occurred as transient solutions (*Danner et al., 2016*).

Unfortunately, there is no experimental evidence for or against brainstem excitation of V3 or inhibition of V0 CINs and LPNs. Either mechanisms could exist and operate in reality. Moreover, they could even coexist and cooperate in multiple ways. This issue can be resolved in future experiments by recording from identified CINs and LPNs and analyzing their response to MLR stimulation.

## Long propriospinal neurons and the organization of interactions between rhythm-generating circuits

The left-right coordination in the present model is mediated by both the local CINs, coupling left and right RGs at cervical and lumbar levels, and the diagonal LPNs, that couple cervical and lumbar RGs. The local CIN interactions (*Figure 2A* and *Figure 3*) were implemented based on earlier experimental studies (*Bellardita and Kiehn, 2015*) and followed our previous computational model (*Danner et al., 2016*). The homolateral and diagonal LPN-mediated pathways (*Figure 2B,C* and *Figure 3*) were incorporated to be consistent with (*Ruder et al., 2016*). The bidirectional homolateral excitatory (from each extensor to the corresponding flexor center, mediated by Shox2 interneurons) and the descending inhibitory (between the flexor centers, mediated by LPNi interneurons) LPN pathways acted synergistically to ensure fore-hind alternation over all frequencies and gaits. The descending diagonal inhibitory LPNs (V0$_D$) together with the local inhibitory V0$_D$ CINs ensured stable quarter-phase lags between homolateral and diagonal RGs during walk. The diagonal ascending and descending excitatory LPNs (V0$_V$) promoted diagonal synchronization and (together with the inhibitory homolateral interactions) supported left-right alternation during trot.

The presence of only cervical-to-lumbar but not lumbar-to-cervical inhibitory LPNs (*Ruder et al., 2016*) and stronger lumbar-to-cervical excitatory influence than cervical-to-lumbar (*Juvin et al., 2005*; *Brockett et al., 2013*) introduced a fore-hind asymmetry. Thus, the dynamics of our model were consistent with left-right but not fore-hind interchange (*Schöner et al., 1990*; *Golubitsky et al., 1999*). The left-right symmetry (because of symmetric LPN and CIN interactions) resulted in stable normalized left-right phase differences of 0.5 (in walk and trot), or 0/1 (in bound), or pairs of steady states symmetric around 0.5 (in gallops). These symmetry properties resulted in the stability of both left and right leading gallops. The fore-hind asymmetry caused the lateral-sequence walk to be the only stable walk, which is the preferred walk of several quadrupeds, including mice, dogs, cats, and humans crawling (*Patrick et al., 2009*; *Bellardita and Kiehn, 2015*; *Righetti et al., 2015*; *Frigon, 2017*), as well as the correct extension phase sequences during gallop and bound. Afferent inputs to LPNs could potentially affect the symmetry properties and result in different stable solutions. For example, cats exhibit diagonal-sequence walk when walking on a split belt treadmill with the fore and hind limbs at a different speed (*Thibaudier et al., 2013*).

## Role of genetically identified spinal interneurons in interlimb coordination

We used our model to simulate the effects of ablating some genetically identified spinal interneurons on the speed-dependent expression and transition of different gaits. The removal of V0$_V$ neurons abolished trot (*Figure 5B*) and only bound could be expressed after the removal of both V0$_V$ and V0$_D$ (*Figure 5D*), which is fully consistent with the experimental data of *Bellardita and Kiehn (2015)*. Also, because in our model V2a neurons mediate input to V0$_V$ neurons (*Figure 2* and *Figure 3*; see *Crone et al., 2008*; *Talpalar et al., 2013*; *Rybak et al., 2015*; *Shevtsova et al., 2015*), removal of V2a had the same effect as removing V0$_V$: transition to gallop and bound at a lower frequency, consistent with *Crone et al. (2009)*.

It is necessary to note that the role of V0$_V$ neurons in the expression and support of trot in the present model significantly differs from our previous model (*Danner et al., 2016*). The previous model had been developed before the new data on LPN organization (*Ruder et al., 2016*) became

available and only included homolateral LPNs and not diagonal ones. Therefore, in that model, the role of V0$_V$ neurons in supporting trot was entirely based on their involvement in support of left-right alternation. In the present model, both V0$_V$ CINs and LPNs support trot, by providing left-right alternation (CINs) and diagonal synchronization (LPNs). The model suggests that removal of one type of these pathways (e.g. the diagonal LPNs, *Figure 5C*) does not fully eliminate trot, but shifts transition to gallop to a lower locomotor frequency. Transition to gallop results from the increasing inhibitory influence of the brainstem drive to V0$_V$ neurons with increasing frequency. The predicted inhibitory influence of brainstem drive (e.g. from MLR) to identified V0$_V$ neurons (both local CINs and LPNs) awaits experimental testing.

## Effect of ablation of the descending cervical-to-lumbar LPN pathways

Simulating deletion of cervical-to-lumbar connections was performed for additional model validation in relation to the recent experimental data: *Ruder et al. (2016)* showed that deletion of these connections, did not affect interlimb coordination at low speeds, disturbed left-right coordination between hind limbs at medium speeds, exhibiting spontaneous switching to left-right synchronization, and caused the same disturbances in left-right coordination between both the fore and hind limbs at high speed. In the model, this deletion resulted in additional steady states with almost synchronous left-right activities of hind RGs at the medium drive values (frequencies) and slow migration of such additional steady states to left-right synchronization of fore RGs (*Figure 6B*).

Introduction of noise led to random transitions between these coexisting stable states, and resulted in spontaneous changes in phase relationships between hind RGs (including occasional synchronization) at medium frequencies and both hind and fore RGs at higher frequencies (*Figure 7*). This qualitatively corresponded to the experimental results on ablation of cervical-to-lumber connections (*Ruder et al., 2016*) and hence provided additional validation to our model.

## Commissural and long propriospinal neurons can serve as main targets for supraspinal and sensory afferent signals controlling limb coordination

Although limbed animals, including quadrupeds express gaits according to their speed, there are many other factors affecting selection of the appropriate gaits, including energetic/metabolic (*Hoyt and Taylor, 1981*), biomechanical (*Alexander and Jayes, 1983*; *Biewener, 1990*; *Farley and Taylor, 1991*; *Fukuoka et al., 2015*), environmental and behavioral factors. For example, animals need to maintain balance when maneuvering in dynamic environments, or when chasing prey or escaping from a predator. As locomotor activities are controlled by networks within the spinal cord, signals affecting interlimb coordination to accommodate these factors can be (a) common or specific supraspinal inputs or drives, reflecting additional visual, auditory, vestibular and different goal-directed factors, and/or (b) proprioceptive and cutaneous afferent feedbacks from the periphery that informs the spinal controller about various biomechanical constraints. The supraspinal control of gait transitions has been confirmed by multiple experiments during fictive or real locomotion controlled by brainstem stimulation (*Orlovskiĭ et al., 1966*; *Shik et al., 1966*; *Shik and Orlovsky, 1976*; *Skinner and Garcia-Rill, 1984*; *Grillner, 1985*; *Atsuta et al., 1990*; *Nicolopoulos-Stournaras and Iles, 2009*). At the same time, there are experimental data confirming the important role that proprioceptive and exteroceptive feedback plays in the control of interlimb coordination and gait (reviewed by *Pearson, 1995*; *Pearson, 2004*; *Frigon, 2017*).

The locomotor gait of quadrupeds is defined by the phase relationships of the movements of the different limbs. Assuming that each limb is controlled by a separate RG, each gait can be represented as a set of phase differences between the rhythmic activities generated by the RGs. In quadrupeds, all left-right interactions within the cervical and lumbar spinal cord, including interactions between left and right RG circuits, are mediated by local CINs, and all diagonal and homolateral interactions between the cervical and lumbar circuits, including interactions between the corresponding RGs, are mediated by LPNs with descending and ascending, diagonal and homolateral projections, respectively. Therefore, the activities of CINs and LPNs located within the four sections of the cord (left and right hemicords at cervical and lumbar levels) explicitly define interactions between the four rhythm-generating circuits (or RGs; *Figure 9*). An integrated pattern of activity of these neurons defines interlimb coordination and gait. Multiple studies confirm that spinal CINs and

LPNs receive excitatory and inhibitory inputs from supraspinal structures (*Skinner et al., 1979*; *Skinner et al., 1980*; *Alstermark et al., 1987*; *Bannatyne et al., 2003*; *Jankowska et al., 2003*; *Jankowska et al., 2005*; *Jankowska et al., 2006*; *Matsuyama et al., 2004*; *Cowley et al., 2008*; *Cowley et al., 2010*; *Jankowska, 2008*; *Szokol et al., 2011*; *Zaporozhets et al., 2011*; *Juvin et al., 2012*; *Mitchell et al., 2016*; *Ruder et al., 2016*), which could explicitly provide speed-dependent gait control as in our model. At the same time, as we showed in our simulations (*Figure 8G–I*), inputs to CINs and LPNs other than supraspinal drives controlling locomotor frequency can also induce gait changes independent of speed. In particular, these inputs can come from sensory afferents. This is consistent with multiple experimental data showing that CINs and LPNs receive direct and indirect input from sensory afferents (*Lloyd and McIntyre, 1948*; *Edgley et al., 2003*; *Bannatyne et al., 2006*; *Jankowska et al., 2009*; *Flynn et al., 2017*). In accord with this is the notion that afferent feedback from each limb projects mostly to the spinal cord section controlling the same limb (*Frigon, 2017*) and hence its influence on spinal circuits and RGs controlling other limbs should be mostly mediated by CINs and LPNs located in this section.

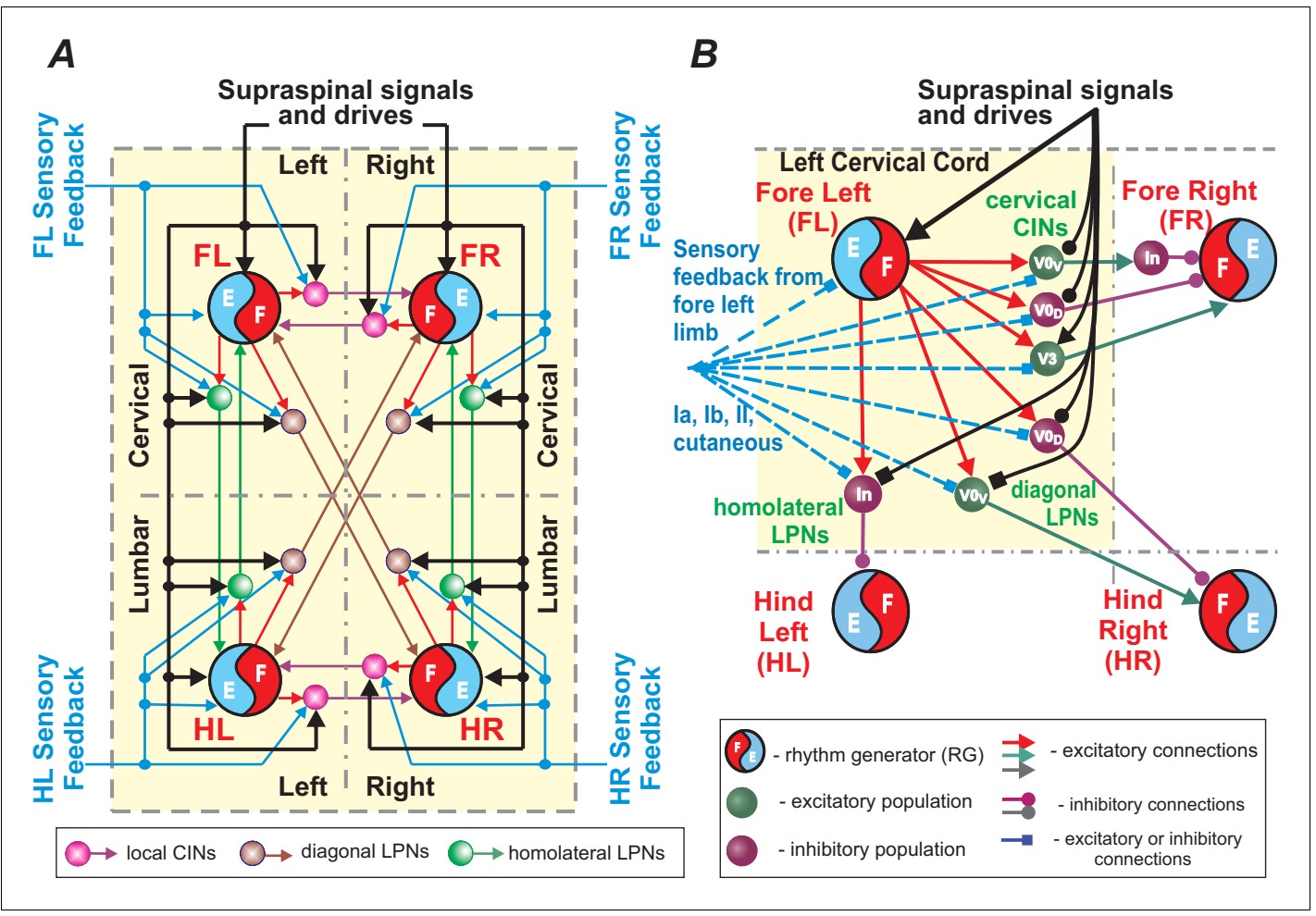

**Figure 9.** Schematic representation of control of limb coordination and gait by different local commissural interneurons (CINs) and long propriospinal neurons (LPNs). (**A**) Circuits controlling four limbs. Each limb is controlled by its own rhythm generator (RG). Local CINs and homolateral and diagonal LPNs couple the RGs and coordinate their activities. Supraspinal signals and sensory feedbacks (directly or also indirectly through dorsal horn interneurons; not shown) provide inputs to the RGs, CINs and LPNs. Inputs to the RGs affect the locomotor frequency (speed), while inputs to CINs and LPNs affect interlimb coordination and gait expression. (**B**) A more detailed representation of local CINs and LPNs that are located in one cord section (here left cervical) and integrate the corresponding intraspinal and supraspinal inputs and sensory information (from the fore left limb) to mediate the effect of this limb's activity on spinal circuits controlling the other three limbs.
DOI: https://doi.org/10.7554/eLife.31050.010

Therefore, we suggest that CINs and LPNs represent the main neural targets for different local/intraspinal, supraspinal, and sensory inputs to control interlimb coordination and adjust gait to various internal and external conditions. Inputs to other neural structures, such as the RG centers, could also alter interlimb coordination by transsynaptically influencing the balance of the CIN and LPN pathways, but would also affect other parameters, such as locomotor frequency. Separate (independent of brainstem drives) signals to CINs and LPNs only (e.g. from sensory afferents) can affect limb coordination and change gait independent of speed (as shown in *Figure 8G–I*). *Figure 9* summarizes the neural circuitry within the spinal cord responsible for interlimb coordination and possible inputs from supraspinal and peripheral afferents. For simplicity, in this schematic, the same CIN and LPN populations receive all intraspinal, supraspinal and afferent inputs. In real cord, the effects of these inputs are rather mediated by separate or overlapping populations of CINs and LPNs, which may interact with each other (e.g. via presynaptic inhibition) competing for their influence on gait expression.

## Model limitations

In this study, we focused only on central interactions, without considering biomechanical constraints and afferent feedbacks from the limbs. Mechanical coupling between the limbs and body can also influence interlimb coordination and interact with the neural control (*Nishikawa et al., 2007*). Moreover, gaits are not only characterized by the phase differences between the limbs but also by a change in kinematics and muscle activation patterns (*Bellardita and Kiehn, 2015*). Investigating these issues requires simulation of limb and body biomechanics in addition to the spinal circuits, which is outside the scope of the current study. We also did not consider spinal circuits operating below the RGs, such as pattern formation networks, circuits related to muscle afferent inputs, reflex circuits including Ia, and Ib interneurons, Renshaw cells and motoneurons (see *Rybak et al., 2006a*, *2006b*; *McCrea and Rybak, 2007*, *2008*; *Zhong et al., 2012*), which play an important role in controlling the limbs. Moreover, how inputs from various supraspinal structures interact with different spinal mechanisms and afferent feedback at different spinal levels in the context of controlling interlimb coordination remains largely unknown. All the above will be the focus of our future investigations.

# Materials and methods

## Neuron model

The model consists of a network of interconnected populations of neurons. Each population was represented by a non-spiking, 'activity-based' model (*Ermentrout, 1994*) and described by

$$C \cdot dV/dt = -I_{\mathrm{NaP}} - I_{\mathrm{L}} - I_{\mathrm{SynE}} - I_{\mathrm{SynI}} - I_{\mathrm{Noise}} \qquad (1)$$

for flexor and extensor centers, and by

$$C \cdot dV/dt = -I_{\mathrm{L}} - I_{\mathrm{SynE}} - I_{\mathrm{SynI}} - I_{\mathrm{Noise}} \qquad (2)$$

for all other populations. In *Equations 1 and 2*, $V$ represents the average membrane potential, $C$ the membrane capacitance, $I_{\mathrm{NaP}}$ the persistent sodium current, $I_{\mathrm{L}}$ the leakage current, $I_{\mathrm{SynE}}$ and $I_{\mathrm{SynI}}$ excitatory and inhibitory synaptic currents, respectively, and $I_{\mathrm{Noise}}$ a noisy current.

The leakage current was described by

$$I_{\mathrm{L}} = g_{\mathrm{L}} \cdot (V - E_{\mathrm{L}}), \qquad (3)$$

where $g_{\mathrm{L}}$ is the leakage conductance and $E_{\mathrm{L}}$ the leakage reversal potential.

Excitatory and inhibitory synaptic currents ($I_{\mathrm{SynE}}$ and $I_{\mathrm{SynI}}$, respectively) of population $i$ were described by:

$$I_{\text{SynE},i} = g_{\text{SynE}} \cdot \left\{ \sum_j \left[ S(w_{ji}) \cdot f(V_j) \right] + D_{\text{E},i} \right\} \cdot (V_i - E_{\text{SynE}}); \tag{4}$$

$$I_{\text{SynI},i} = g_{\text{SynI}} \cdot \left\{ \sum_j \left[ S(-w_{ji}) \cdot f(V_j) \right] + D_{\text{I},i} \right\} \cdot (V_i - E_{\text{SynI}}); \tag{5}$$

where $g_{\text{SynE}}$ and $g_{\text{SynI}}$ are synaptic conductances, and $E_{\text{SynE}}$ and $E_{\text{SynI}}$ the reversal potentials of the excitatory and inhibitory synapses; $w_{ji}$ is the synaptic weight from population $j$ to $i$ ($w_{ji}>0$ for excitatory connections and $w_{ji}<0$ for inhibitory connections); and function $S$ is defined as

$$S(x) = \begin{cases} x, & \text{if } x \geq 0 \\ 0, & \text{otherwise} \end{cases}. \tag{6}$$

The output function $f(V)$ translates $V$ into the integrated population activity or neural output and was defined by the linear piecewise function:

$$f(V) = \begin{cases} 0, & \text{if } V<V_{\text{thr}} \\ (V - V_{\text{thr}})/(V_{\text{max}} - V_{\text{thr}}), & \text{if } V_{\text{thr}} \leq V<V_{\text{max}} \\ 1, & \text{if } V \geq V_{\text{max}} \end{cases}. \tag{7}$$

Excitatory $D_{\text{E},i}$ and inhibitory $D_{\text{I},i}$ drives to population $i$ were modeled as a linear function of the free parameter $\alpha$:

$$D_{\{\text{E},\text{I}\},i}(\alpha) = m_{\{\text{E},\text{I}\},i} \cdot \alpha + b_{\{\text{E},\text{I}\},i}, \tag{8}$$

where $m_{\{\text{E},\text{I}\},i}$ is the slope, and $b_{\{\text{E},\text{I}\},i}$ the intercept. $\alpha$ represents the value of the variable brainstem drive. If not otherwise specified, $m_{\{\text{E},\text{I}\},i}$ and $b_{\{\text{E},\text{I}\},i}$ were set to 0 (lack of drive input to the corresponding neuron population).

The persistent sodium current in the flexor and extensor centers of the RG was described by

$$I_{\text{NaP}} = \bar{g}_{\text{NaP}} \cdot m \cdot h \cdot (V - E_{\text{Na}}), \tag{9}$$

where $\bar{g}_{\text{NaP}}$ is the maximal conductance, $m$ and $h$ activation and inactivation variables of $I_{\text{NaP}}$, respectively, and $E_{\text{Na}}$ the sodium reversal potential. Activation of the persistent sodium current was considered instant and was modeled by

$$m(V) = \{1 + \exp[(V - V_{1/2,m})/k_m]\}^{-1} \tag{10}$$

and slow inactivation obeyed the following differential equation

$$\tau_h(V) \cdot dh/dt = h_\infty(V) - h \tag{11}$$

with

$$h_\infty(V) = \{1 + \exp[(V - V_{1/2,h})/k_h]\}^{-1}; \tag{12}$$
$$\tau_h(V) = \tau_0 + (\tau_{\text{max}} - \tau_o)/\cosh[(V - V_{1/2,\tau})/k_\tau]. \tag{13}$$

In **Equations 10–13**, $V_{1/2}$ and $k$ represent half-voltage and slope of the corresponding variable, respectively, and $\tau_0$ and $\tau_{\text{max}}$ are the baseline and maximum of inactivation time constant $\tau_h$, respectively.

The noisy current $I_{\text{Noise}}$ was described as an Ornstein-Uhlenbeck process

$$dI_{\text{Noise}}/dt = -I_{\text{Noise}}/\tau_{\text{Noise}} + \sigma_{\text{Noise}}\sqrt{2/\tau_{\text{Noise}}} \cdot \xi_i(t) \tag{14}$$

where $\tau_{\text{Noise}}$ is the time constant, $\sigma_{\text{Noise}}$ the standard deviation or strength of the noise, and $\xi_i(t)$ (in $1/\sqrt{\text{s}}$) was the population-specific normalized Gaussian noise.

## Model parameters

The model parameters were adapted from our previous model (*Danner et al., 2016*): $C = 10$ pF; $g_L = 4.5$ nS for RG centers and $g_L = 2.8$ nS for all other neurons; $\bar{g}_{NaP} = 4.5$ nS; $g_{SynE} = g_{SynI} = 10.0$ nS; $E_L = -62.5$ mV for RG centers and $E_L = -60$ mV for all other neurons; $E_{Na} = 50$ mV; $E_{SynE} = -10$ mV; $E_{SynI} = -75$ mV; $V_{thr} = -50$ mV; $V_{max} = 0$ mV; $V_{1/2,m} = -40$ mV; $V_{1/2,h} = -45$ mV; $k_m = -6$ mV; $k_h = 4$ mV; $\tau_0 = 80$ ms; $\tau_{max} = 160$ ms; $V_{1/2,\tau} = -35$ mV; $k_\tau = 15$ mV; and $\tau_{Noise} = 10$ ms.

Following drive parameters were used: $m_E = 0.0$ and $b_E = 0.1$ for the extensor centers; $m_E = 0.1$ and $b_E = 0.0$ for the flexor centers; $m_I = 0.75$ and $b_I = 0.0$ for all V0$_D$ CINs; and $m_I = 0.15$ and $b_I = 0.0$ for the local homologous V0$_V$ CINs. Thus, the extensor centers received constant drive (independent of $\alpha$) and the flexor centers and V0 CINs received variable drive whose changes were proportional to $\alpha$. The connection weights are listed in *Table 1*. Weights of connections within the RGs were adapted from our previous model (*Danner et al., 2016*) and other weights were selected within their operating ranges and tuned to allow gait transitions to occur at experimentally observed locomotor frequencies (*Bellardita and Kiehn, 2015*). To test the robustness of the model, we simultaneously varied all connection weights; each weight was multiplied by a normally distributed random number with a mean of 1 and standard deviation $\sigma_p$ of 0.01, 0.02, 0.05, and 0.10. For each $\sigma_p$, 100 random models were created and bifurcation diagrams were calculated. With $\sigma_p \leq 0.02$ all randomized models retained all stable regimes and their sequential transitions with changes of $\alpha$, at $\sigma_p = 0.05$ 17% and at $\sigma_p = 0.10$ 40% of the models lost some stable solutions (gaits such as bound or trot). Thus, the final model represents a *coarse* system allowing parameter variations without dramatic (qualitative) changes in behavior. To simulate deletion of particular populations, their outputs were set to 0.

**Table 1.** Connection weights.

| Source | Target ($w_{ij}$) |
| --- | --- |
| | Within girdle and side of the cord |
| RG-F | i-Ini-F (0.40), i-V0D (0.70), i-V2a (1.00), i-V3 (0.35), i-V2a-diag (0.50) |
| f-RG-F | i-Ini-Hom (0.70), i-V0D-diag (0.50) |
| RG-E | i-Ini-E (0.40), i-CINi (0.40), i-Sh2-Hom (0.50) |
| Ini-F | i-RG-E (–1.00) |
| Ini-E | i-RG-F (–0.08) |
| V2a | i-V0V (1.00) |
| V2a-diag | i-V0V-diag (0.90) |
| IniV0V | i-RG-F (–0.07) |
| | Between left and right homologous circuits |
| V0D | c-RG-F (–0.07) |
| V0V | c-IniV0V (0.60) |
| V3 | c-RG-F (0.03) |
| CINi | c-RG-F (–0.03) |
| | Between fore and hind homolateral circuits |
| f-Ini-Hom | h-RG-F (–0.01) |
| f-Sh2-Hom | h-RG-F (0.01) |
| h-Sh2-Hom | f-RG-F (0.125) |
| | Between diagonal circuits |
| f-V0D-diag | dh-RG-F (–0.075) |
| f-V0V-diag | dh-RG-F (0.02) |
| h-V0V-diag | df-RG-F (0.065) |

i-, ipsilateral; c-, contralateral; f-, fore; h-, hind; d, diagonal. CINi, inhibitory commissural interneurons. Ini, regular inhibitory interneurons. RG-F, flexor center; RG-E, extensor center.

DOI: https://doi.org/10.7554/eLife.31050.011

## Computer simulations

The set of differential equations were solved using the odeint (*Ahnert et al., 2011*) implementation of the Runge-Kutta-Fehlberg 7–8 variable step-size solver of the boost C++ library (version 1.63.0). $I_{\mathrm{Noise}}$ was calculated before the simulation with the Forward Euler method and for a fixed step size of 1 ms. The custom C++ code was compiled with clang 800.0.42.1 (Apple LLVM 8.0.0) for macOS 10.12.3. Simulation results were analyzed using Matlab 2016b. Source code and Matlab scripts for all simulations are available in ModelDB (*McDougal et al., 2017*) at http://modeldb.yale.edu/234101.

## Data analysis

Activities of the flexor and extensor center were used to assess the model behavior. A RG was considered as being in flexion when $f(V)$ of its flexor center was greater or equal to 0.1, otherwise it was considered as being in extension. The locomotor period was defined as the duration between two consecutive onsets of the left hind flexor center; the frequency as the reciprocal of the period; extension and flexion phase durations as the duration between onset and offset of extension and flexion, respectively. Normalized phase differences were calculated as the delay between the onsets of the extension phases of a pair of rhythm generators divided by the locomotor period. Four normalized phase differences were calculated: hind left-right (right hind – left hind), fore left-right (right fore – left fore), homolateral (left fore – left hind) and diagonal (right fore – left hind). Gaits were operationally defined based on the phase differences (see *Table 2*).

## Analysis of model performance

To identify stable solutions of the model, bifurcation diagrams were built for the four normalized phase differences. To this end, $\alpha$ was increased from 0.0 to 1.05 and then decreased back to 0.0 in 1000 equally spaced steps (step size of 0.00105). At each step, the simulation was run for 10 s and the final state was used as the initial condition for the next step. Circular statistics were evaluated for the last five locomotor cycles. At each step, the circular standard deviation of the normalized phase differences of the last five steps was smaller than 0.001. Thus, the mean phase differences can be regarded as the stable solutions. The sequential increase and decrease of $\alpha$ was performed to uncover regions where multiple states are stable. An additional simulation run with stepwise change of $\alpha$ in the opposite direction was initialized when a discrete change of phase differences between two $\alpha$-values occurred, to uncover potentially missed stable trajectories. Initial conditions were randomized and multiple runs were evaluated. $\sigma_{\mathrm{Noise}}$ was set to 0.005 pA, making noisy currents several orders of magnitude smaller than any other current. The noise did not affect the model behavior other than to ensure that the system did not remain on an unstable trajectory.

To model step-to-step variability as presented in decabper, simulations with increased noisy currents ($\sigma_{\mathrm{Noise}} = 1.75$ pA) were performed. The free parameter $\alpha$ was set to 0.3, 0.6, and 0.75. At each value, the simulation was run for 1000 s. Phase differences were calculated and partitioned into three bins (cf. *Figure 7*), depending on their appropriateness for trot and to ensure comparability with experimental data (*Ruder et al., 2016*). The counts per bin were then divided by the total number of locomotor cycles and multiplied by 100.

To model dynamics of gait transitions, extension phases and the instantaneous frequency are illustrate before and after an abrupt change in $\alpha$-value or additionally introduced drives. To this end,

**Table 2.** Operational definition of gaits.

| Gait | Left-right hind | Normalized phase differences | |
| --- | --- | --- | --- |
| | | Homolateral | Diagonal |
| Walk* | [0.25,0.75] | [0.1,0.4) and (0.6,0.9] | (0.1,0.4] and [0.6,0.9) |
| Trot | [0.25,0.75] | [0.25,0.75] | [0.0,0.1] and [0.9,1.0] |
| Gallop | (0.025,0.25] and [0.75,0.975) | [0.25,0.75] | [0.25,0.75] |
| Bound | [0.0,0.025] and [0.975,1.0] | [0.25,0.75] | [0.25,0.75] |

*Classification of walk additionally required longer extension than flexion phase durations.

DOI: https://doi.org/10.7554/eLife.31050.012

the model was initialized with a predefined $\alpha$-value and ten seconds of settling period were allowed before the abrupt parameter change. Parameters used are specified in the legend of *Figure 8*.

---

# Additional information

## Funding

| Funder | Grant reference number | Author |
| --- | --- | --- |
| National Institutes of Health | R01 NS081713 | Ilya A Rybak |
| National Institutes of Health | R01 NS090919 | Ilya A Rybak |
| National Institutes of Health | R01 NS095366 | Natalia A Shevtsova |

The funders had no role in study design, data collection and interpretation, or the decision to submit the work for publication.

## Author contributions

Simon M Danner, Conceptualization, Software, Formal analysis, Investigation, Visualization, Methodology, Writing—original draft, Writing—review and editing; Natalia A Shevtsova, Investigation, Methodology, Writing—review and editing; Alain Frigon, Conceptualization, Writing—review and editing; Ilya A Rybak, Conceptualization, Supervision, Funding acquisition, Investigation, Visualization, Methodology, Project administration, Writing—review and editing

## Author ORCIDs

Simon M Danner http://orcid.org/0000-0002-4642-7064
Ilya A Rybak http://orcid.org/0000-0003-3461-349X

## Decision letter and Author response

Decision letter https://doi.org/10.7554/eLife.31050.015
Author response https://doi.org/10.7554/eLife.31050.016

---

# Additional files

## Supplementary files

• Transparent reporting form
DOI: https://doi.org/10.7554/eLife.31050.013

---

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
