## [Decision Letter]

Thank you for submitting your article "Long propriospinal neurons and gait expression in quadrupeds" for consideration by *eLife*. Your article has been reviewed by three peer reviewers, one of whom, Ronald L Calabrese (Reviewer #1), is a member of our Board of Reviewing Editors, and the evaluation has been overseen by Eve Marder as the Senior Editor. The following individuals involved in review of your submission have agreed to reveal their identity: Stephen Soffe (Reviewer #2); Rob Brownstone (Reviewer #3).

The reviewers have discussed the reviews with one another and the Reviewing Editor has drafted this decision to help you prepare a revised submission.

Summary:

This modeling study uses the latest available data on spinal locomotor networks and behavior in mice and constructs a model to explore possible mechanisms of gait transitions. The model focuses on interneurons that interconnect limb CPGs both locally within the fore or hind limb enlargement and between the limb centers. It emphasizes the introduction of contralaterally projecting long propriospinal neurons (LPNs) as previous models from the same group did not include these recently described LPNs, but only short projecting commissural interneurons (CINs) and ipsilaterally projecting LPNs. Certain populations of LPNs and CINs are subject to changing descending excitatory drive that targets the flexor half-center of the limb CPGs. The model captures well speed-dependent gait expression based on this descending drive. The model is further constrained by testing its ability to capture changes in gait caused by genetic ablation of specific groups of spinal interneurons. In particular, it captures newly discovered disruption of hindlimb coordination following ablation of descending LPNs. They further show that inputs to CINs and LPNs can affect interlimb coordination and change gait independent of speed. The model makes the strong prediction that LPN and CINs interneurons represent the main targets for supraspinal and sensory afferent signals adjusting gait.

Spinal networks controlling locomotion are a very active area of research and this model organizes the vast amount of available data on genetic identification, connectivity, genetic ablation, and behavior and presents testable scheme for the organization of interlimb coordination. We are impressed by the model and the way it has evolved from a previous version, and found the paper a very interesting read. It is an advance on the previous published version because it incorporates new data that endows the model with new explanatory and predictive power. As such, it should be of wide interest.

The work is carefully done and the illustrations are beautiful and helpful as well as presenting the necessary data.

Essential revisions:

1) Rather than being a fundamentally new approach, this paper presents an advance in a model that has been evolving: for example recently presented in their 2016 (J. Physiol) paper. This new version of the model presents a more sophisticated connectivity based on new physiological/behavioral data, i.e. it follow the availability of more detailed evidence on homolateral and crossed pathways between fore and hind-limb centers (Ruder et al., 2016). Interestingly, the 2016 version already presented a model within which control of descending drive to specific points in the spinal network produced changes in locomotor speed and concomitant gait changes. So what does the addition of further connectivity add? It’s necessary to make clearer the relationship between the two models. Do the diagonal connections add something fundamentally different or are they there to fine tune coordination? It seems that they 'improve' some transitions between gaits (e.g. discussed briefly around paragraph four of the Discussion) but do they fill an important functional gap in the 2016 model? The Introduction should be a more thorough job of putting the model in the context of what is known in the field and what the major issues are.

Does the similarity in the capability of the 2016 and new model show a weakness of relatively unconstrained modelling if you can get similar functionality out of quite different levels of coupling? Some discussion along these lines would seem appropriate.

2) The paper provides a clear story of how coordination could work through a balance of different influences. However we are concerned about how robust the model is in terms of choices of connection weights. The authors acknowledge, quite rightly, that direct experimental evidence is seriously lacking. How were the various strengths determined and how critical are they? A further addition to the previous model (as far as we can tell) is the element of descending inhibitory drive to V0_V_s and V0_D_s; is there experimental evidence to support this? It would be helpful to provide a clearer picture in this paper of some of the key history and limitations of the model.

3) A major concern relates to the descending drive. The authors have justified why they have set it up as is (because it works). Whether there is a necessity for projection to the extensor centers (which receive "relatively high drive") is not clear (as it produces only tonic activity in them). It's not until the second paragraph of the Discussion that it becomes clear that there are 2 different (independent?) descending drives – clarity about this is important. Is there a physiological basis to this? It gets less clear in the third paragraph of the Discussion. Could the "drive" to extensors not be accomplished by simply an increase in excitability of the target neurons? Modulators, for example? Most importantly, though: α is poorly defined. Furthermore, at times α and locomotor frequency are used interchangeably. Is the relationship between the two the same in all conditions (the various knockouts)? As the drive forms the crux of the study, it is important to be crystal clear about it, how the values were assigned, what they mean, what is the physiological basis for these values, etc. Can Figure 2 be used to illustrate what happens when α is changed? Finally, given the relative paucity in knowledge about descending drive, explicitly stating the predictions of its nature in the discussion would be helpful. A corollary concern relates to sensitivity of the model to changes in α. For example, in Figure 8 how dependent are the transitions in G-I on the exact α value chosen? Why are 3 significant figures used to specify this parameter; is extreme precision needed?

4) Variability, noise, and Figure 7. This is very difficult to follow. It is not clear what this means in terms of physiology – where is the increasing noise coming from? And given that the noise is increased to the degree that it is, can the reader assume that there is a high degree of tolerance of these circuits to noise? Or do they normally operate near this high limit and thus need to be very well tuned to prevent susceptibility to noise? Figure 7 is difficult to interpret.

5) Afferent feedback is first mentioned early in the Introduction (in an odd sentence stating that the feedback projects to the limbs?). It then arises later in the Discussion. This topic appears tangential to the manuscript and not fully explored. The authors might want to consider leaving this for a subsequent manuscript devoted to the topic, exploring how sensory stimulation can alter circuit function. In any case, they might want to consider removing this topic from the current manuscript to gain on focus and clarity. (This would also mean Figure 9 isn't needed.) (Furthermore, even sensory input confined to fibres terminating in the dorsal horn (not included in this model) can affect gait. Presumably this would be via indirect pathways.) In fact, the authors say in subsection “Model limitations” that they aren't considering afferent input, so why are they?

6) It might be interesting from a comparative biology point of view to consider scaling in the discussion. What happens as you move from mice to cats to horses, for example? Could there be a "simple" evolutionary process (in terms of these circuits) that changes the relationship between α and locomotor frequency? And speaking of horses, there has been recent work on Dmrt3, which is expressed in a subset of spinal interneurons. Dmrt3 mutations lead to alterations in interlimb coordination (studies replicated in mice). How do these neurons (dI6?) fit into the picture? I came to this by looking at Figure 6D2, which is very close to an even, 4-beat gait as seen in the Icelandic horse tölt. What would be needed for this circuit to produce a tölt? Or to produce a pace, in which the two left limbs alternate with the two right limbs?

7) The model. How were the values in Table 1 determined? Is there a physiological basis for them, or were they derived iteratively? Either way is okay, but we should know where the numbers come from. Furthermore, how robust is the model to changes in these values? Would the degree of robustness tell us anything about the circuits or their development?

8) Several qualitative comparisons to experimental data are made. These should to the extent that is possible be made quantitative, e.g., the data of Figure 7 vs. C1-E1.

9) Not all readers interested in locomotor CPGs are well versed in bifurcation diagrams. While we concur with the authors that such diagrams convey a lot of information and insight succinctly, the crux is that they must be understood. The bifurcation diagrams are not explained adequately in the text. Help lead the reader through them by explicit reference to specific curves. In particular, describe what is multistability and how it appears in the diagrams. Be clear about the hysteresis and explain how it is uncovered and visualized in the diagrams. Help the reader through these diagrams (one example thoroughly explained will do); your work deserves to be understood.

10) Materials and methods are not fully adequate. Give more details about the model, e.g., how many cells are in a population? Did all cells in a population have exactly the same parameters, inputs, noise? Beef up the sections on Analysis of model performance and Data analysis. Make the complete model and all the simulations available on ModelDB or similar site and on DRYAD or similar sites respectively. The model code etc. MUST be posted in a way that others have direct and complete access to it.

---

## [Author Response]

Essential revisions:1) Rather than being a fundamentally new approach, this paper presents an advance in a model that has been evolving: for example recently presented in their 2016 (J. Physiol) paper. This new version of the model presents a more sophisticated connectivity based on new physiological/behavioral data, i.e. it follow the availability of more detailed evidence on homolateral and crossed pathways between fore and hind-limb centers (Ruder et al., 2016). Interestingly, the 2016 version already presented a model within which control of descending drive to specific points in the spinal network produced changes in locomotor speed and concomitant gait changes. So what does the addition of further connectivity add? It’s necessary to make clearer the relationship between the two models. Do the diagonal connections add something fundamentally different or are they there to fine tune coordination? It seems that they 'improve' some transitions between gaits (e.g. discussed briefly around paragraph four of the Discussion) but do they fill an important functional gap in the 2016 model? The Introduction should be a more thorough job of putting the model in the context of what is known in the field and what the major issues are.Does the similarity in the capability of the 2016 and new model show a weakness of relatively unconstrained modelling if you can get similar functionality out of quite different levels of coupling? Some discussion along these lines would seem appropriate.

We substantially revised the Introduction to clarify the state of knowledge in the field in general and in connection with differences between our previous (Danner et al., 2016) and present model.

Concerning the differences between the previous (Danner et al., 2016) and the present models, there are the two major differences. The first difference concerned the organization of descending (cervical-to-lumbar) and ascending (lumbar-to-cervical) connections mediated by LPNs. The previous model had been developed before the data of Ruder et al. became available. Therefore, at that time, we just tried to find and a minimal solution for LPN pathways organization that would allow the model to reproduce the major gaits and transitions between them. The LPN connectome proposed in that model included only homolateral inhibitory interactions between the cervical and lumbar RGs and did not include any diagonal interactions. The study of Ruder et al., 2016 did not confirm the proposed connectome. Hence, our task in the present study was to find another solution that, on one hand, would allow reproducing the major gaits and their transitions, and, on the other hand, would be consistent with the current knowledge on the organization of LPN pathways, particularly with the recent Ruder et al., 2016 study. Such a solution has been found and, we believe, represents a step toward better understanding of spinal circuit organization. Incorporating diagonal LPN connections (and removal of ascending inhibitory connections) in the present model was primarily made to fit Ruder et al. data, and may not represent a fundamental difference except the fact that these revised connections allowed the existence of stable gallop, which in the previous model could only occur as a transient regime. We have highlighted the difference between the models concerning gallops in the last paragraph of the section “Trot and the transition to gallop and bound: diagonal V0_V_ LPNs stabilize trot” of the Results.

The second difference between the previous (Danner et al., 2016) and the present model was in the organization of supraspinal inputs to CINs and LPNs provided the speed-dependent gait transitions. Importantly, in both models we have suggested that gait transitions are caused by brainstem drives acting on these neurons and changing the balance between the pathways promoting left-right synchronization and left-right alternation. In the previous model, the transition from alternating (trot) to synchronized gaits (gallop and bound) resulted from excitatory drive to local V3 CINs, promoting left-right synchronization at both cervical and lumbar levels. In the current model, the same conceptual idea was implemented by increasing inhibitory influence of the brainstem drive on local V0 CINs (V0_D_ and V0_V_) and diagonal V0_D_ LPNs involved in left-right alternation. Both implementations of the same idea can exist and operate in reality. Moreover, they can coexist and cooperate in multiple ways. Unfortunately, at the current insufficiency of experimental data, we cannot say which of these solutions is more realistic. However, we do not think that suggesting alternative solutions represents a weakness of the model; we would rather consider this as an advantage. Moreover, we now not only suggest the possible alternative versions, but also propose experiments (e.g., recording the responses of different identified CINs and LPNs to MLR stimulation) that can resolve this issue and show which of the suggested versions is more realistic and closer to reality. This issue is now additionally discussed at the very end of the section “Control of locomotor speed and speed-dependent gait transitions” of the Discussion.

2) The paper provides a clear story of how coordination could work through a balance of different influences. However we are concerned about how robust the model is in terms of choices of connection weights. The authors acknowledge, quite rightly, that direct experimental evidence is seriously lacking. How were the various strengths determined and how critical are they? A further addition to the previous model (as far as we can tell) is the element of descending inhibitory drive to V0_V_s and V0_D_s; is there experimental evidence to support this? It would be helpful to provide a clearer picture in this paper of some of the key history and limitations of the model.

The weights of connections were selected (within their operating ranges) to allow the changes in the balance (and hence in the corresponding gait transitions) to occur at the values of α that corresponded to the experimentally measured values of frequency at which these gait transitions occurred. Hence, relative changes of these weights can change/shift the corresponding points of gait transitions (values of α and transition frequencies). Yet, the model represents a coarse system and allows parameter variations without dramatic (qualitative) changes in behavior. This is now mentioned at the end of section “Model parameters” of Materials and methods.

Inhibition of V0 neurons by brainstem drive as a mechanism for gait transitions does not have experimental support so far. This is our suggestion and model prediction. In the previous model (Danner et al., 2016), we had a different idea that the increasing brainstem drive activates the excitatory V3 CINs. Unfortunately, there is no experimental evidence for or against brainstem excitation of V3 or inhibition of V0 CINs and LPNs. Either, or both, mechanism can operate in reality. This issue can be resolved in future experiments by recording from genetically identified CINs and LPNs and analyzing their response to MLR stimulation. We state this now in the Discussion section (last paragraph of “Control of locomotor speed and speed-dependent gait transitions”).

3) A major concern relates to the descending drive. The authors have justified why they have set it up as is (because it works). Whether there is a necessity for projection to the extensor centers (which receive "relatively high drive") is not clear (as it produces only tonic activity in them). It's not until the second paragraph of the Discussion that it becomes clear that there are 2 different (independent?) descending drives – clarity about this is important. Is there a physiological basis to this? It gets less clear in the third paragraph of the Discussion.

This comment concerns several different issues and we respond to the individual issues separately. First of all, the idea that increasing brainstem drive causes increasing locomotor frequency (speed) and speed-dependent gait transitions was based on multiple experimental data showing that MLR stimulation in decerebrate cats and rats induce locomotor activity, whose frequency increased with increasing intensity of stimulation (Orlovskii et al., 1966; Shik et al., 1966; Shik and Orlovsky, 1976; Skinner and Garcia-Rill, 1984; Grillner, 1985; Atsuta et al., 1990; Nicolopoulos-Stournaras and Iles, 2009). In many of these studies, such an increase in locomotor frequency was accompanied by sequential changes in gait from walk to trot, gallop and bound. This is described in our paper in the third paragraph of the section “Control of locomotor speed and speed-dependent gait transition” in the Discussion.

An independent high drive to the extensor centers represents a very important issue. The possible origins of this drive had been only shortly discussed in the Discussion (second paragraph; i.e., separate descending drive through reticulospinal or vestibulospinal tracts and/or inputs from cutaneous afferents and load receptors from extensor muscles). This issue is related to the flexion-dominated concept of locomotor CPG organization (Pearson and Duysens, 1976; Duysens, 1977; Zhong et al., 2012; Duysens et al., 2013; Machado et al., 2015), which was used in our previous models (Molkov et al., 2015; Shevtsova et al., 2015; Danner et al., 2016; reviewed in detail in Rybak et al., 2015, and Ausborn et al., 2017). The idea is that the extensor centers can independently generate intrinsic rhythmic bursting under certain conditions (similar to the flexor centers, e.g., see Hägglund et al., 2013). However, as in the previous models, we suggested that under normal conditions, they receive high excitation, keeping them out of intrinsic bursting mode, and hence they normally operate in a regime of tonic activity and exhibit rhythmic bursting due to the rhythmic inhibition from the intrinsically oscillating flexor centers. This was a basis for considering separate drives to flexor and extensor centers.

We agree that we have not adequately explained the two different drives. We have now included an explanation and clarification of the extensor and flexor drives with the corresponding references to the first paragraph of the section "The model" in Results and made corresponding changes in second paragraph from the end of this section.

Could the "drive" to extensors not be accomplished by simply an increase in excitability of the target neurons? Modulators, for example?

It could, and as soon as extensors are excited enough to operate in the regime of tonic activity (if isolated). Therefore, it does not matter whether their excitation is produced by a separate excitatory drive (through synaptic activation) or by modulators. However, the former scenario looks more realistic, since it is more difficult to imagine that modulators act selectively only on extensor-related RG neurons.

Most importantly, though: α is poorly defined. Furthermore, at times α and locomotor frequency are used interchangeably. Is the relationship between the two the same in all conditions (the various knockouts)? As the drive forms the crux of the study, it is important to be crystal clear about it, how the values were assigned, what they mean, what is the physiological basis for these values, etc.

In the first sentence of the section “Control of locomotor frequency and gait by brainstem drive” in the Results we stated that parameter α “defined brainstem drive to flexor centers, CINs, and LPNs” with reference to Materials and methods (where the dependence of brainstem drive(s) on α is clearly defied by equation 8 with all parameters specified in this chapter). Then, we stated “the model generated oscillations when parameter α was changed from 0 to 1.05. Within this range an increase of α led to an increase in locomotor frequency from 2 to 12 Hz (Figure 4, top diagram) […].” Also, Figure 4 (top diagram) clearly shows the direct relationship between α and locomotor frequency generated by the model, including the relationships for values of α when gait changes. The frequency of oscillation in this and previous models was fully defined by the brainstem drive to flexor centers and were almost independent of the phasic interactions between RGs mediated by CINs and LPNs, which were much weaker and could only affect phase relationships between the RGs, but not the oscillation frequency or amplitude. This was previously discussed by Rybak et al., 2015 and has been now mentioned in first paragraph of the section “The Model” in Results. Because of this, the dependences of frequency and phase durations on α, shown in Figure 4, were almost unchanged with removal of any CIN and LPN types.

The only physiological basis for the relationship between the α (conditionally representing brainstem or MLR drive) and the locomotor frequency is the data showing that MLR stimulation in decerebrate cats and rats induce locomotor activity whose frequency increased with increasing intensity of stimulation (Orlovskii et al., 1966; Shik et al., 1966; Shik and Orlovsky, 1976; Skinner and Garcia-Rill, 1984; Grillner, 1985; Atsuta et al., 1990; Nicolopoulos-Stournaras and Iles, 2009), which we mentioned in several places through the paper, but the exact dependence has never been studied experimentally, neither in intact, nor in genetically transformed animals.

Can Figure 2 be used to illustrate what happens when α is changed? Finally, given the relative paucity in knowledge about descending drive, explicitly stating the predictions of its nature in the discussion would be helpful. A corollary concern relates to sensitivity of the model to changes in α. For example, in Figure 8 how dependent are the transitions in G-I on the exact α value chosen? Why are 3 significant figures used to specify this parameter; is extreme precision needed?

Figure 2 shows (physical) connectivity in the model. We are not sure how we can illustrate what happens when α changes. We have changed the color of the brainstem drive to the extensor centers in Figure 2 and Figure 3 to make clear that this drive is separate from the drive to flexor centers, and that the drive to the flexors and to CINs/LPNs have the same origin. The current knowledge about the brainstem drive is now described in the revised Introduction (second paragraph from the end of Introduction) and in the second paragraph of the Discussion. In the Discussion (including the revised last paragraph) we describe our predictions concerning the effects of brainstem drive to CINs and LPNs. Based on the above, any change in α (drive to the flexor centers) will result in the corresponding change in the locomotor frequency and, in some critical points, can also change gait as shown in Figure 4.

The three examples shown in Figure 8 were chosen to show three different gait transitions. For example, Figure 8 shows a transition from trot to gallop and bound at an α value of 0.5 when additional inhibitory drives were applied to V0_V_ CINs and LPNs. This α value was chosen because it is in the middle of the range of α where trot is expressed (see Figure 5). The α-values in Figure 8 have more significant digits because gallop is expressed in a much narrower α-range than trot (see Figure 5).

We were specifically interested in showing the dynamics gait transitions so they can be related to the examples of Bellardita and Kiehn, 2015 and to the model when α was changed. The three figures (diagrams) in Figure 8 were selected to reproduce similar changes as exemplified in the paper of Bellardita and Kiehn (2015, Figure 3). The correspondence here was only qualitative, the exact tuning of α was not necessary.

4) Variability, noise, and Figure 7. This is very difficult to follow. It is not clear what this means in terms of physiology – where is the increasing noise coming from? And given that the noise is increased to the degree that it is, can the reader assume that there is a high degree of tolerance of these circuits to noise? Or do they normally operate near this high limit and thus need to be very well tuned to prevent susceptibility to noise? Figure 7 is difficult to interpret.

This issue requires explanation. It does not matter where this noise came from and what is the level of noise (if the latter is within a reasonable range). The noise (representing natural stochastic processes or multiple signals of unknown origin) is always present in any real system. What is important here is how many steady states (regimes) simultaneously exist (co-exist) in the system. If the system has only one steady regime, it operates close to this regime, and noise can produce a small deviation around it. However, if a nonlinear system has two (or more) steady states (regimes) that coexist (which indicates bi-/multistability), then noise allows the system to spontaneously “jump” from one steady regime to another. In such case, during some time intervals, the system operates in, or close to, one steady regime and during some other time intervals, operates in or around another steady regime(s). This is a general property of multistable dynamic systems. In this connection, the level of the noise just defines how often such jumps happen. Therefore, the noise (including natural noise or uncorrelated inputs in real system) allows distinguishing the above two situations.

In the section “Deletion of descending (cervical-to-lumbar) LPNs affects left-right coordination” of the Results, with reference to Figure 6, we described that deletion of the cervical-to-lumbar LPNs in our model resulted in the emergence of new stable states when α (and frequency) was increasing. Specifically, at low brainstem drive (α) values (or low locomotor frequencies) walk and then trot remained the only stable state, but at medium drive (α) values (medium frequencies) both steady trot and steady gallop coexisted at the same drive (α) values. These changes in steady regimes become obvious with including noise as shown in Figure 7. An increased noise resulted in spontaneous transitions between two steady regimes (seen in Figure 6 and described above): one characterized by left-right alternation (phase differences around 0.5, specific for trot, and the other characterized by left-right relationships corresponding to gallop (see Figure 7, indicated by dashed square, and Figure 7, orange and red shaded areas). These modeling results were qualitatively very similar to the experimental data of Ruder et al., 2016 and hence provide further validation of our model. We have significantly revised the whole section “Noise causes high step-to-step variability after deletion of cervical-to-lumbar LPNs”, including descriptions related to Figure 7, to make them easier to follow for the reader.

5) Afferent feedback is first mentioned early in the Introduction (in an odd sentence stating that the feedback projects to the limbs?). It then arises later in the Discussion. This topic appears tangential to the manuscript and not fully explored. The authors might want to consider leaving this for a subsequent manuscript devoted to the topic, exploring how sensory stimulation can alter circuit function. In any case, they might want to consider removing this topic from the current manuscript to gain on focus and clarity. (This would also mean Figure 9 isn't needed.) (Furthermore, even sensory input confined to fibres terminating in the dorsal horn (not included in this model) can affect gait. Presumably this would be via indirect pathways.) In fact, the authors say in subsection “Model limitations” that they aren't considering afferent input, so why are they?

We agree that the topic of afferent inputs has not been explicitly explored and this is now mentioned in the section “Model limitations” of the Discussion. Our main conclusion of the paper is that we believe that CINs and LPNs are the main target for signals that control or influence interlimb coordination. In Figure 8 we showed that additional signals (other than descending brainstem drive that also controls frequency) to CINs and LPNs can cause changes in interlimb coordination and gait. Such control, could originate from some supraspinal centers. However, there is extensive evidence of projections from sensory afferents to CINs and LPNs. Therefore, we suggest that sensory feedback from each limb can also contribute to limb coordination and gait expression through CINs and LPNs. Based on this, we would prefer to keep the discussion of afferent feedback and Figure 9 in the paper. We updated the text to make clear that afferent feedback has not been explicitly simulated in our model. We also clarified the relationship between this conclusion and the results in Figure 8. Appropriate changes have been made in the last two paragraphs of the section “Commissural and long propriospinal neurons can serve as main targets for supraspinal and sensory afferent signals controlling limb coordination” in Discussion.

The potential role of dorsal horn interneurons in mediating sensory inputs has been mentioned in the legend of Figure 9.

6) It might be interesting from a comparative biology point of view to consider scaling in the discussion. What happens as you move from mice to cats to horses, for example? Could there be a "simple" evolutionary process (in terms of these circuits) that changes the relationship between α and locomotor frequency? And speaking of horses, there has been recent work on Dmrt3, which is expressed in a subset of spinal interneurons. Dmrt3 mutations lead to alterations in interlimb coordination (studies replicated in mice). How do these neurons (dI6?) fit into the picture? I came to this by looking at Figure 6D2, which is very close to an even, 4-beat gait as seen in the Icelandic horse tölt. What would be needed for this circuit to produce a tölt? Or to produce a pace, in which the two left limbs alternate with the two right limbs?

In this study, we tried to be careful and to use and reproduce data that are mostly related to mice. We think that scaling or extending our model to some other animals, such as horses, would be beyond the reasonable degree of speculation. Moreover, concerning locomotion, the specific differences may be connected not with evolution per se, but rather with the size, weight, kinematics, and the relationships of these characteristics to the limited maximal muscle force and other muscle characteristics. That can explain why, for example, not all results obtained in mice are reproducible in primates.

Concerning the inclusion of dI6 neurons in the model circuitry: in our previous paper (Danner et al., 2016), we suggested that the inhibitory commissural neurons CINi (see Figure 3), promoting left-right synchronization, could be a subset of dI6 neurons, but this suggestion has no support so far. In fact, we could not find a clear, specific formulation of the role of dI6 neurons in locomotion/gait control in mice, or of the effect of their selective ablation, besides the general statement that they project to both sides of the cord and “play multiple roles during locomotor activity” (Andersson et al., 2012; Dyck et al., 2012; Greiner et al., 2017). Therefore, we have been so far unable to formulate task for their modeling.

Furthermore, the presence or absence of the DMRT3 mutation influences the horse’s ability to pace (Andersson et al., 2012). However, as far as we know mice do not express pace (Bellardita and Kiehn 2015; Lemieux et al., 2015). This further complicates modeling of dI6 neurons and their involvement in the control of locomotion. Pace could potentially be reproduced in our model by including additional circuits that promote synchronization of homolateral rhythm generators (e.g., mutual excitation of the flexor centers) or alternation of diagonal rhythm generators (e.g., mutual inhibition of diagonal flexor centers) together with a mechanism that suppresses the present pathways promoting homolateral alternation and diagonal synchronization.

7) The model. How were the values in Table 1 determined? Is there a physiological basis for them, or were they derived iteratively? Either way is okay, but we should know where the numbers come from. Furthermore, how robust is the model to changes in these values? Would the degree of robustness tell us anything about the circuits or their development?

Table 1 specifies the weights of connections between neural populations in the model. The connection weights for the RG circuits were adapted from the previous model (Danner et al., 2016). All other weights have been selected within their operating ranges and tuned to reproduce the necessary behaviors of the model under considered conditions (experimentally described changes in locomotion/gait after genetic removal of particular neuron types). There is no specific (explicit) experimental data about connectivity patterns of these neurons or strength of their connections.

In the previous model (Danner et al., 2016), we tuned the parameters of RG units and their interactions to allow them to generate locomotor oscillation within the physiological range of locomotor frequencies with realistic phase durations under control of external (brainstem drive). Changes in these parameters can affect values and range of generated frequencies and phase durations. These parameters were adapted in the present model. Connection weights of the LPN and CIN pathways were tuned to make sure that gait transitions occur at the proper values of α (frequency) in order to fit corresponding experimental data. Changes in these weights can lead to gait transition at wrong frequencies or (with more dramatic changes in weights) suppress the expression of particular gaits. However, in general, the model represents a*coarse system* allowing parameter variations without dramatic (qualitative) changes in behavior. The process of parameter selection has been described with more details in the “Model parameters” section in Materials and methods.8) Several qualitative comparisons to experimental data are made. These should to the extent that is possible be made quantitative, e.g., the data of Figure 7 vs. C1-E1.

With the account of multiple simplifications made in the model, we believe, that only qualitative comparisons make sense here. This specifically concerns Figure 7, when our goal was only to show the speed-dependent increase of errors in limb coordination after ablation of cervical-to-lumbar LPN connections. The quantitative matching was not expected there and, in our opinion, was not really necessary.

9) Not all readers interested in locomotor CPGs are well versed in bifurcation diagrams. While we concur with the authors that such diagrams convey a lot of information and insight succinctly, the crux is that they must be understood. The bifurcation diagrams are not explained adequately in the text. Help lead the reader through them by explicit reference to specific curves. In particular, describe what is multistability and how it appears in the diagrams. Be clear about the hysteresis and explain how it is uncovered and visualized in the diagrams. Help the reader through these diagrams (one example thoroughly explained will do); your work deserves to be understood.

We thank the reviewers for this comment. We have inserted detailed explanations on how to read bifurcation diagrams in the very beginning of the section “Walk and the transition to trot: diagonal V0_D_ LPNs ensure stable walk”. Furthermore, we now explicitly reference to the specific curves when we explain the bifurcation diagrams of the intact model.

10) Materials and methods are not fully adequate. Give more details about the model, e.g., how many cells are in a population? Did all cells in a population have exactly the same parameters, inputs, noise? Beef up the sections on Analysis of model performance and Data analysis. Make the complete model and all the simulations available on ModelDB or similar site and on DRYAD or similar sites respectively. The model code etc. MUST be posted in a way that others have direct and complete access to it.

This is probably misunderstanding. As we described in Materials and methods, “each population was represented by a non-spiking, ‘activity-based’ model (Ermentrout, 1994).” So in contrast to many our previous models (e.g. Rybak et al., 2006a,b; 2013; Zhong et al., 2012; Shevtsova et al., 2015; Shevtsova and Rybak, 2015), where we explicitly model populations of neurons described in the Hodgkin-Huxley style, we here followed our other models (Molkov et al., 2015; Danner et al., 2016), in which each population was modeled as a single unit using a non-spiking, ‘activity-based’ model (Ermentrout, 1994). The model equations are clearly described in Materials and methods (Equations 1-7) and the use of non-spiking population models had been additionally mentioned in the very beginning of the Results.

We deposited the full source code and model files on ModelDB. This code can be used to run all simulations presented in the paper. Additional Matlab scripts are provided that run these simulations, process the data, and create plots. Following line has been added to the “Computer simulations” section in the Materials and methods: “Source code and Matlab scripts for all simulations are available in ModelDB (McDougal et al., 2017) at http://modeldb.yale.edu/234101.” For now, the source code can be accessed with “1234” as a passcode (without quotation marks). ModelDB only allows publication of the model files after publication. This passcode won’t be necessary once the paper has been published.